# Methodological concerns underlying a lack of evidence for cultural heterogeneity in the replication of psychological effects
Robin Schimmelpfennig [1,8] ✉, Rachel Spicer [2,8] ✉, Cindel J. M. White [3], Will Gervais[4], Ara Norenzayan[5], Steven Heine[5], Joseph Henrich[6] & Michael Muthukrishna [2,7] ✉

The multi-site replication study, Many Labs 2, concluded that sample location and setting did not substantially affect the replicability of findings. Here, we examine theoretical and methodological considerations for a subset of the analyses, namely exploratory tests of heterogeneity in the replicability of studies between "WEIRD and less-WEIRD cultures". We conducted a review of literature citing the study, a re-examination of the existing cultural variability, a power stimulation for detecting cultural heterogeneity, and re-analyses of the original exploratory tests. Findings indicate little cultural variability and low power to detect cultural heterogeneity effects in the Many Labs 2 data, yet the literature review indicates the study is cited regarding the moderating role of culture. Our reanalysis of the data found that using different operationalizations of culture slightly increased effect sizes but did not substantially alter the conclusions of Many Labs 2. Future studies of cultural heterogeneity can be improved with theoretical consideration of which effects and which cultures are likely to show variation as well as a priori methodological planning for appropriate operationalizations of culture and sufficient power to detect effects.

Many psychological findings do not replicate well[1,2], likely due to methodological limitations and a lack of robust theory[3–5]. In the past decade, we have learned much about the reasons behind these replication failures, which have prompted methodological reform and the development of 'best practices'[4,6,7]. In a separate vein, there has been growing awareness that our knowledge of human behavior is heavily skewed by an empirical dataset overwhelmingly composed of people from Western, Educated, Industrialized, Rich, and Democratic (WEIRD) societies[8–10]. However, a large and growing body of contemporary cultural evolutionary theory and empirical data reveals that humans are a cultural species, evolved to be contextually and culturally embedded decision-makers. That is, people learn from those around them how to think, feel, and reason[11–13], resulting in culturally shaped experiences and potentially psychological differences across populations. This makes it problematic to build a behavioral science from any single population.

While both problems, replication failures, and homogenous (biased) sampling, have received widespread attention, surprisingly, little is still known about the potential links between the two, that is, the role of population diversity in the replicability and the effect sizes of psychological findings. Generalizing beyond WEIRD cultural psychology requires that published studies are replicated with cross-cultural samples to demonstrate the robustness and limitations of psychological effects[14].

With this goal in mind, a high-profile research project attempted to address the moderating role of population, site, and setting variability in replication failures[15]. In the large-scale, multi-site project Many Labs 2 (ML2), Klein et al.[15] ran 28 classic and contemporary research studies distributed over 125 sample sites, comprising 15,305 participants in 36 countries. They found that 14/28 effects (50%) showed a statistically significant effect ($p < 0.0001$) in the same direction as the original study (15 effects replicated with the common threshold of $p < 0.05$). In a pre-registered design, ML2 mainly focused on testing whether the 28 included effects varied across different contexts (e.g., paper/pencil vs. computer-based, different sample site). They found little heterogeneity in replicability level, representing the main finding of ML2. An additional exploratory analysis investigated cultural variation as a potential explanation for heterogeneity and non-replication across samples. Specifically, the researchers tested

[1]Faculty of Business and Economics, University of Lausanne, Lausanne, Switzerland. [2]Department of Psychological and Behavioural Science, London School of Economics and Political Science, London, United Kingdom. [3]Department of Psychology, York University, Toronto, ON, Canada. [4]Centre for Culture and Evolution, Psychology, Brunel University, London, United Kingdom. [5]Department of Psychology, University of British Columbia, Vancouver, BC, Canada. [6]Department of Human Evolutionary Biology, Harvard University, Cambridge, MA, USA. [7]Canadian Institute for Advanced Research (CIFAR), Toronto, ON, Canada. [8]These authors contributed equally: Robin Schimmelpfennig, Rachel Spicer ✉e-mail: robin.schimmelpfennig@unil.ch; r.a.spicer@lse.ac.uk; m.muthukrishna@lse.ac.uk

whether a binary "*WEIRDness*" scale moderated each effect. In the following, when we talk about ML2, we are mainly referring to this subset of the project, and we refer to the "*ML2-WEIRDness*" scale to distinguish between the scale and the original backronym[9]. The cultural background was determined for each sample site based on the country where the sample was situated. The *ML2-WEIRDness* score was calculated by decomposing the backronym into five constituent letters, associating those letters with the underlying term, finding a way to measure that term ("Westerness"), aggregating these five scores, and taking a mean split to partition *WEIRD* from non-*WEIRD* countries. The heterogeneity of samples, comparing Klein et al.'s classifications of *WEIRD* vs non-*WEIRD*, was calculated using the Q, tau, and $I^2$ measures[16]. The authors found few moderating effects of the *ML2-WEIRDness* scale. After correcting for multiple comparisons, Klein et al. found evidence that in 3 of 28 replicated studies, the effects were significantly moderated by *ML2-WEIRDness*[17–19].

The project represents a laudable effort, specifically concerning the main goal, which was exploring heterogeneity in the replicability of findings across samples and settings, tackling an important concern in the psychological sciences[20]. However, we want to raise some potential issues related to the explorative ex-post analysis of the moderating effect of culture. We only draw on ML2 here as an illustrative example, but other projects, potentially also some that we have co-authored, may share some of these issues. Importantly, the authors of ML2 transparently acknowledge that this part of the analysis was exploratory. We think engaging and possibly improving this approach is still important. Even if the focus on culture did not represent the main target of ML2, the authors chose to feature the results prominently, for example, in the summary of the study's results in the abstract, asserting that: "*Exploratory comparisons revealed little heterogeneity between Western, educated, industrialized, rich, and democratic (WEIRD) cultures and less WEIRD cultures (i.e., cultures with relatively high and low ML2-WEIRDness scores, respectively)*." Consequently, the results have received increased attention in the literature and are cited as evidence for the role of cross-cultural differences in the replicability of psychological phenomena.

We identify five theoretical and methodological considerations in the way ML2 tested for the moderating role of culture variability. Our concerns with the ML2's explorative ex-post cultural moderation analysis are as follows:

Our first concern is the importance of theory in the selection of studies and sample sites for replication. Since the cultural moderation analysis was exploratory, Klein et al.[15] did not theoretically motivate the selection of studies based on cultural differences. They did not provide theoretically grounded (ex-post) predictions regarding which psychological effects will and will not generalize cross-culturally. In light of the theoretical literature on cross-cultural variability in psychology[13,21,22], such explorative testing not motivated by theory can be problematic. For example, ML2 included the study by Huang et al.[17], who experimentally explored cultural differences in metaphoric associations of living in the north or the south of a city. However, the ML2 project then expanded the list of the sampled populations beyond the original populations (US and Hong Kong) without considering the explanation for why these specific cardinal directions might lead to different metaphorical associations across populations, especially because there is no reason to expect that north-south economic differences would be geographically universal (e.g., Canada is unlikely to fit this pattern). While Klein et al. did conduct subset analyses, including only participants from the US and Hong Kong respectively, for they included the entire sample for their main analysis. Indeed, in some cases, one might 'replicate' a successful test of a theory without replicating the precise empirical patterns in the data because a proper theory may predict different results across locations. Despite considering this issue in their pre-registration, it was not obvious from the available statistical code how and why this north-south dimensionality was expanded to other countries in the analysis. The main motivation for replicating the study across the world without an a priori hypothesis may have been that the original studies did implicitly generalize their findings to all humanity.

In other cases, the initial cultural context of a study was altered without theoretical reasoning about expected variation in the new context. For example, Norenzayan et al.'s study[19] found cultural differences in a highly standardized sample of US college students who were very similar (i.e., matched based on cognitive abilities and education) except for their cultural background. At the same time, ML2 expanded the test to include variation across countries without considering the author's original reasoning in standardizing the design[23]. It is possible that the effects were selected based on theoretical foundations, but we could not find these stated in the article or the pre-registration.

Similarly, the selection of sample sites does not seem to have been guided by hypotheses about meaningful cultural variation. This can be explained by the ex-post nature of the analysis, which may constrain the conclusions we can draw from the results. Many Labs 2 made sampling choices by convenience to assess the role of sampling heterogeneity. Crucially, using conveniently available samples is not a harmless choice because, given the cultural background of the lead authors and the structure of their social networks, this can produce biased sets of populations. Of course, psychologists are familiar with the representativeness heuristic, which in this situation will lead many readers to implicitly assume that this convenience sample is roughly equivalent to a representative or random sample. It is not, as we will demonstrate. Our second concern is sampling WEIRD people around the world. The ML2 subject pool consisted of participants who are predominantly US-based and overwhelmingly WEIRD (see Fig. 1 for sample composition of ML2's Slate 1 and Slate 2). Indeed, only a fraction of the participants were obtained from non-Western populations. 39% of all participants were sampled from the US – arguably the WEIRDest of all countries[9].

We can be more precise about the extent of this cultural homogeneity. Cultural fixation ($CF_{ST}$) provides a continuous measure of cultural similarity between groups across a range of cultural traits[10]. As the United States is a country at the extreme end of the WEIRD spectrum and has the largest number of sampling sites, a country's cultural distance from the US can be used as an index of how WEIRD that sample is.

Furthermore, despite cross-*country* variation in sample sites, the sites were all based at universities and Amazon MTurk, known to oversample from high SES populations relative to the world population[24]. Thus, the samples are likely to be skewed towards participants who are high SES, highly educated, and digitally literate, and therefore, much more likely to be at the "WEIRD" end of the distribution within each society. Just as sharing a religious affiliation predicts cultural similarity among people living in different countries[25], sharing a high socioeconomic status and participating in WEIRD institutions, such as higher education, may drive cultural similarities across nations[26]. As a point of reference, we remind readers that there are over 700 million adults in the world who've never been to school at all (twice the U.S. population). Of course, we recognize the practical and financial constraints that researchers face when recruiting globally diverse samples; nevertheless, the strength of inferences about generalizability across populations is proportional to the extent and scope of diversity captured in sampling choices.

Our third concern is the cultural background of participants is not necessarily that of the sample country. The issue of sampling WEIRD people is further exacerbated by calculating the *ML2-WEIRDness* score for each sample site based on the country of origin of that site, irrespective of the original country of origin of participants. Clustering by the source country of the sample site rather than by the individuals' origins conceals potential psychological variation, including the possible migration background of participants. Culture should not be equated with site country, and thus, both approaches, identifying culture by sample site and home country, should be handled with caution (e.g., how long has a person lived in their home country? At what age did they arrive?). Nonetheless, since the original claims were made based on using a sample country as a proxy for culture and these are the data available, in our re-analysis below, we also used the sample site country and home country as an alternative approach.

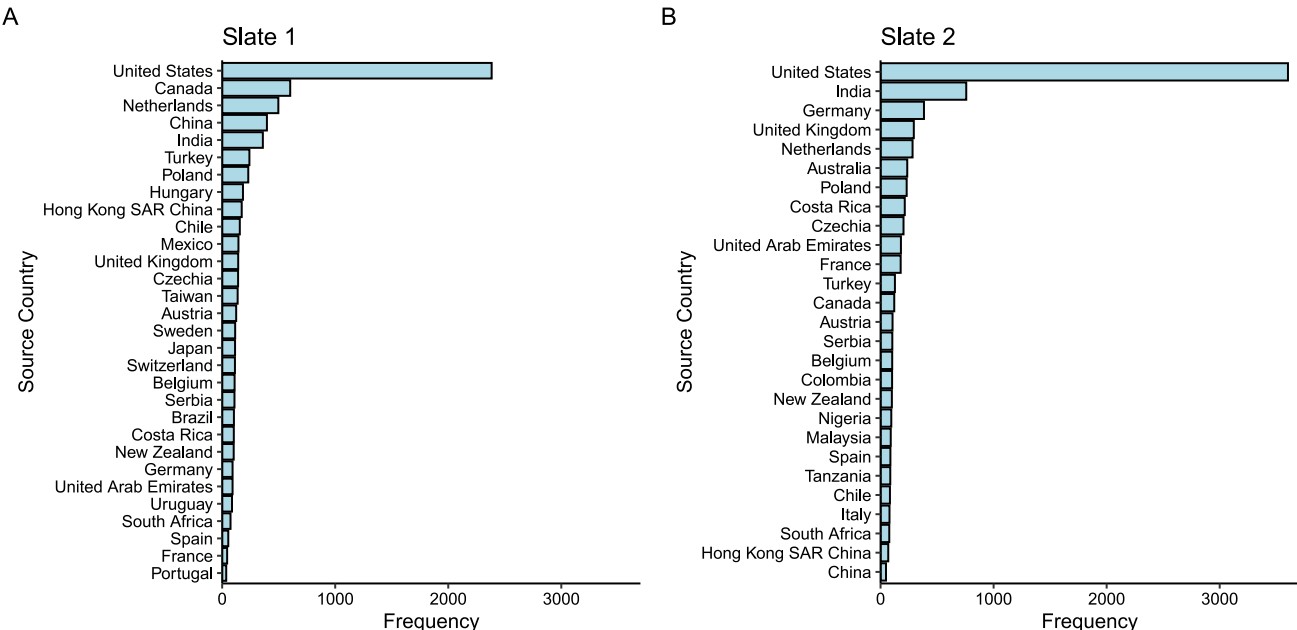

**Fig. 1 | The frequency of participants from each source country included in ML2.** Figure **A** shows the frequency of Slate 1 and Fig. **B** shows the frequency of Slate 2. The minimum number of participants included in any one sample is 36. This figure is based on publicly available ML2 data, and it has been previously published in a Commentary[34].

Our fourth concern is assessing cultural moderation by decomposing the backronym (WEIRD), a rhetorical device. Klein et al. assessed cultural moderation by decomposing the letters of the WEIRD backronym. This division of countries has been copied in several other papers[27,28]. We are unaware of any conceptual, empirical, or theoretical justification for this. Indeed, this is inconsistent with the original formulation of the WEIRD people problem[9]. The catchy backronym captures some aspects of the regions and demographics overrepresented in the psychological record – namely, being Western, Educated, Industrialized, American, and high SES (often students) – but it hardly captures the defining characteristics or mechanisms driving differences between these societies. WEIRD was designed as a consciousness-raising device reminding experimental behavioral scientists about psychological diversity[8], not as a theoretical operationalization of the explanatory concept. In their introduction, Henrich et. al. explain:

"We emphasize that our presentation of telescoping contrasts is only a rhetorical approach guided by the nature of the available data. It should not be taken as capturing any unidimensional continuum or suggesting any single theoretical explanation for the variation[9]."

Henrich et al. were explicit that the WEIRD backronym is not intended to embody or summarize any key theoretical or conceptual factors important for explaining global psychological variation. This would turn a mnemonic conscious-raising device into a theoretical construct. Of course, one could take the backronym as a motivation for a purely inductive, empirically-driven approach based on cultural differences among populations–without bestowing any theoretical import on the backronym's letters–to generate such a measure.

Our fifth concern is the mean split of the WEIRD scale. A further issue is that the authors ascribed binary codes (1 and 0) to sample sites based on a mean split and thus summed up all WEIRD, especially less-WEIRD samples, under the same category. Such dichotomies necessarily conceal important variability. A more precise, continuous coding of cultural distance between samples would be more appropriate. These shortcomings are obvious when one looks at the ML2 coding of their samples, which results in rather surprising binary *ML2-WEIRDness* values. For example, the sample site at the American University of Sharjah in the UAE with its gold-plated pillars is coded as non-rich; Chile was coded as categorically the same as Germany and Sweden but categorically different from near-neighbors

Spanish-speaking Costa Rica and Uruguay. South Africa was coded categorically like China and India, but it was categorically different from other Commonwealth states such as Australia and New Zealand. This lack of face validity also indicates the adverse effects of missing theory about culture and population variability[3].

Importantly, other cultural distance measures could be used to assess the effect of sampling variability. For example, Muthukrishna et al.[10] developed a tool to measure the degree of similarity between different groups' cultural values, beliefs, and practices. This measure provides an empirically-driven way to quantify worldwide cultural variability by measuring the overall cultural distance between any two countries for which data is available (Muthukrishna et al. used the 2005 and 2010 waves of the World Values Survey[10,29]). Other alternative measures of cultural heterogeneity that may better capture population variability include tightness and looseness[30], and individualistic and collectivistic cultures[31].

However, improved measures of cultural heterogeneity and theory-driven selection of samples and effects are not the only imperatives for large multi-site projects investigating the moderating role of culture with heterogeneity tests. Another crucial and often overlooked issue that would be important for future studies to address is sample size.

With these observations and caveats out in front, we developed an alternative approach to analyze the ML2 data that addresses three of our stated considerations about the: (3) the cultural background of participants not necessarily being that of the sample country, (4) *ML2-WEIRDness variable*, and (5) Mean Split. Unfortunately, neither an inductive measure for cultural differences nor a more accurate approach to identifying cultural background on the individual level can address considerations (1) the importance of theory in the selection of studies and sample sites for replication and (2) sampling mostly WEIRD people around the world. Moreover, as ML2 found that many of the results of the individual studies do not replicate and thus have no evidentiary value, we should not expect these null findings to vary cross-culturally. So, at best, an improved re-analysis will extract the variation in this available data while recognizing that these samples may lack sufficient variation and statistical power. As previously stated in our pre-registration, the results of our re-analysis should thus not be perceived as a final verdict on the moderating role of culture in the given effects. Answering such a question about cultural heterogeneity would require data from a project specifically designed for the task. To inform

future studies in this realm, we chose to illustrate how some of the problems we state can be operationalized in an improved and theory-driven methodological approach.

To calibrate expectations for what conclusions about heterogeneity in replicability one could reasonably expect to emerge from ML2's design choices, and thus also for a possible re-analysis of their results, we conducted a series of simulation studies. The overarching goal was to closely mirror the design, sampling, and analyses of a multi-site study like ML2 (regarding selected effects, samples included, and analyses conducted) while manipulating the extent of cultural influence: the degree to which culture moderates effect sizes. In making cultural influence an exogenously determined variable, we can obtain an answer regarding the degrees of cultural heterogeneity that a setup such as ML2 is well-designed to detect. While the simulations mimic the characteristics of ML2, they could equally be adapted to simulate power for other multi-lab studies (see ref. 32 for power calculation for meta-analyses).

ML2's moderation analysis "*[…]revealed little heterogeneity between Western, educated, industrialized, rich, and democratic (WEIRD) cultures and less WEIRD cultures […]*"[15]. This result can be interpreted in at least two ways. One possibility is that ML2 was well-powered, the analyses provide a strong test of cultural heterogeneity, and the results show a small influence of culture, probably due to low rates of cultural heterogeneity. Another possibility is that culture has a large influence, and results indicate that the design and implementation of ML2 are poorly suited for detecting such differences. In essence, our simulation is trying to understand which of the two scenarios (little cultural influence, but well-measured; unknown-to-high cultural influence, but poorly measured due to issues in statistical power) is more likely to be true, given a design like ML2. Importantly, we focused on cultural variation for the heterogeneity tests in the simulation. Still, the setup and the implications can equally apply to other types of heterogeneities in samples and settings.

When discussing statistical power, an important caveat applies to the level at which statistical power is calculated. Whenever Klein et al. discuss statistical power, they do so in the context of detecting the main effects within each of the 28 studies they replicated. Less attention is put on the effective statistical power of the heterogeneity tests (e.g., heterogeneity in setting or country) that are the inferential backbone of their paper[33]. However, without calculating the statistical power of the heterogeneity tests, their analysis might be unable to deliver evidence of heterogeneity even if it's present.

To illustrate the above considerations numerically and show their potential influence on conclusions drawn from the results, we performed a series of simulation studies, simulating a multi-site study like ML2. Our simulations may help provide context for interpreting the (mostly) insignificant heterogeneity tests and inform intuitions about whether ML2 reflects little heterogeneity that's been well-measured or an unknown degree of heterogeneity in a design that may be severely underpowered to detect it. Therefore, the simulation exercise results are important guidance for researchers designing similar multi-site studies in the future.

Our initial simulation results focused on the power to detect different levels of cultural influence, given a single typical effect size. However, analytic performance can vary across both degrees of cultural influence and effect sizes. After all, ML2 included effect sizes ranging from practically nonexistent to quite large. Second, we investigated the effect of heterogeneity at a range of effect sizes for the reference country, the USA. In addition to running simulations at the level of a single study, we also tried to simulate the entire ML2 setup with several studies at a project level. To assess the impact of ML2 on the literature we performed a literature search, coding how many articles cited ML2 in the context of cultural differences, moderation by culture or WEIRD samples.

Here, we offer three contributions.

1. We raise several considerations about the ML2 study design's ability to document the moderating role of culture in psychological phenomena. The points include (1) the selection of studies and sample sites for replication, (2) sampling mostly WEIRD people around the world, (3) identifying participants' cultural background by the country where the samples came from, (4) treating the WEIRD backronym as a theory by decomposing it into a *ML2-WEIRDness* scale, and (5) using a mean split of that *ML2-WEIRDness* variable. Importantly, these points are all relevant for future cross-cultural multi-site studies that assess whether population variability moderates the replicability and size of psychological effects.

2. We bolster the implications of ML2's sampling decisions by simulating an ML2-like environment and assessing the degree to which there is sufficient power to test the moderating role of culture on replicability. Given that ML2 found little heterogeneity between *WEIRD* and *less WEIRD* cultures, we aimed to understand whether these results from the study were well designed to study little heterogeneity or whether there was unknown-to-high heterogeneity that was not measured due to power issues. The results of these simulations have implications for other multi-site studies and sources of heterogeneity other than culture.

3. We synthesize the implications of the methodological problems, the simulation approach to detect statistical power, and the pre-registered re-analysis of ML2 to offer a set of guidelines and recommendations for more theoretically motivated, high-powered multi-site investigations of cultural differences in the future. We show how some of the concerns we observe in ML2 can be solved with an improved methodological approach in future studies. We illustrate this alternative methodological approach by re-analyzing ML2 data using a pre-registered multi-step protocol (https://osf.io/6exr4).

## Methods
### Literature search
We conducted a short literature search of all published papers that cite the ML2 study to demonstrate this influence in the literature. The literature search was not preregistered. Our search in the Web of Science included all research articles that cited the Many Labs 2 project (ML2)[15] since its publication https://www.webofscience.com/wos/woscc/summary/59ff78e8-9883-4a90-9be4-866bdab20949e8acedbe/date-descending/1. The search, conducted on September 18th 2023, yielded a sample of 409 articles. 6 articles were either not accessible or did not cite ML2, which left us with a final sample of 403 articles. Two coders were trained in the coding framework, and both coded all articles. A third coder then resolved potential misalignment in codings. The process was as follows: both coders went through each of the manuscripts individually, searched for the citation statement (i.e., the sentence of the paragraph in with the paper cited ML2), and copied the citation statement into the codebook. This could mean that a paper had one or several coding events. As the next step, the coders ascribed the citation statement with one of the four citation categories.

1. Citation statements were coded as *Replicability*, when they made general remarks about replicability, or replicability problems in psychological research.

2. Citation statements were coded as *Moderation* when they discussed the moderation/heterogeneity of effects or replicability.

3. Citation statements were coded as *Culture*, when they made any reference to cultural differences, moderation by culture, WEIRD samples. This was the main code of interest.

4. All other citation statements were summarized under the coding category *Other*.

Importantly, as many papers cited Klein et al.[15] several times, the coders also recorded several citation statements and several code categories for those papers. That is, a paper could have cited Klein et al. both, for the moderation of replicability, but also for the moderation by culture more specifically.

### Simulating the moderating role of culture
To begin with, we created a simulation environment in which multiple studies can be run on simulated psychological effects across different countries. The simulations were not preregistered. Sample countries in the

**Table 1 | Descriptions of variables and parameters for the simulation model**

| Variables | Role | Data generation |
|---|---|---|
| $CFst$ | Models cultural differences across countries. | Sample countries in the simulation were randomly drawn in proportion to their representation in ML2. To operationalize cultural variation across sample countries and participants, each country was assigned an *ML2-WEIRDness* score. |
| *ML2- WEIRDness* | Copies ML2's measure of cultural variation and serves to benchmark how well ML2 was set up to detect cultural heterogeneity | Each country was assigned an *ML2-WEIRDness* score. We then performed their same mean split into *ML2-WEIRDer* and less *ML2-WEIRD* countries |
| $d\_i$ | Models the effect size for any given country i. It is used to calculate cultural heterogeneity. | This is calculated based on the combination of parameters and variables as specified in Eq. (1). |
| **Parameters** | **Role** | **Conditions** |
| Cultural Influence | Manipulates the influence that cultural differences ($CFst$) have on differences in effect sizes. | 0: No Influence<br>0.5: Moderate Influence<br>1: Strong Influence |
| $d\_USA$ | The effect size for the selected reference country, the USA, and as per Eq. (1), scales the level of cultural heterogeneity. | The exact distribution of values depends on the simulation, but this is mostly based on effect sizes in psychological studies (e.g., between 0.05 and 0.7) or effect sizes measured in ML2. |

simulation were randomly drawn in proportion to their representation in ML2 (see ; e.g., USA samples were more likely to be included than samples in Uruguay or the United Arab Emirates). To operationalize cultural variation across sample countries and participants, each country was assigned an *ML2-WEIRDness* score. We then performed their same mean split into *ML2-WEIRDer* and less *ML2-WEIRD* countries and meta-analytically conducted heterogeneity tests exactly as they did, using $Q$, tau, and $I^2$ indices[16]. We also calculated the statistical power for these tests. Therefore, we simulate ML2's broader heterogeneity test of sample sites and the *ML2-WEIRDness* variable. The precise variables used are described in Table 1.

Simulations always include assumptions, and we strove to model all assumptions both transparently and quite generously. The representation of countries and effect sizes is directly mapped onto ML2's design. As shown in Eq. (1), to manipulate the moderating role of culture in the study, we model cultural differences (captured by $CF_{ST}$[10]) between countries and the *influence* these cultural differences had (captured by the variable *cultural influence*). We set the USA as the reference country for effect sizes, as it is by far the most overrepresented in ML2. We modeled heterogeneity by having the true effect size within each country diverge from the USA effect size by an amount related to $CF_{ST}$ and the given cultural influence. Each country's (i) effect size ($d_i$) would thus include the effect size of the chosen reference country, USA ($d_{USA}$), adjusted by the country's cultural distance from the USA ($CF_{ST\_i}$) multiplied by a constant that reflected the degree of cultural influence (*Cultural Influence*) in a given simulation. So, in summary, the level of cultural heterogeneity depends on the cultural differences (modeled by $CF_{ST}$) and the manipulated level of cultural influence. However, as Eq. (1) shows, the level of heterogeneity, which is calculated by drawing on the absolute differences in effect sizes across countries, is also conditional on the modeled effect size in the baseline country, the USA.

$$d_i = \underbrace{d_{USA}}_{\text{Effect size USA}} * \left( 1 - \left( \underbrace{CF_{ST\_i} * \frac{\text{Cultural Influence}}{CF_{ST\_max}}}_{\text{Cultural Moderation}} \right) \right) \quad (1)$$

We simulate three levels of Cultural Influence: no influence, moderate influence, and strong influence (see Fig. 2).

*No influence* (Cultural Influence = 0) refers to a scenario where all countries have the same effect size regardless of cultural differences. Simulations with no influence thus had all countries drawn from the same reference effect size ($d_{USA}$), ignoring any presence of cultural differences ($CF_{ST\_i}$). That is, in the *no influence* condition, each country's $CF_{ST}$ is multiplied by zero, entirely leveling our slate of countries to the same

effective effect size for all effects (e.g., a $d_{USA} = 0.5$ effect in the USA would also be a $d_{CN} = 0.5$ in China – see Fig. 2).

*Moderate influence* (Cultural Influence = 0.5) refers to a scenario where the most culturally distant countries have effects half as large as the USA. Simulations at moderate influence thus represented the most culturally distant countries from the USA as having effect sizes *half as large as* those observed in the USA (e.g., $d_{USA} = 0.5$ in the USA would be $d_{CN} = 0.25$ in China).

*Strong influence* (Cultural Influence = 1) is a scenario where effects entirely attenuate in the most culturally distant countries. Simulations at strong influence thus represented effects fully disappearing in countries the most culturally distant from the USA (e.g., a $d_{USA} = 0.5$ in the USA would be a $d_{CN} = 0$ in China).

We simulated a series of social psychological effects, with multiple studies run across countries, with effect sizes (dUSA) randomly drawn from those observed in ML2. We could then pool results across simulated slates of many studies to see which levels of simulated cultural heterogeneity would be expected to produce overall rates similar to those observed in ML2. In this simulation, we induce variation via cultural differences, but we can equally think about that in terms of heterogeneity in settings more broadly.

In summary, we draw on this simulation setup and run a series of three simulation studies:

1. In the first simulation, we simulate a single study on a psychological effect across countries and estimate the *power to detect heterogeneity in the case of a given effect size typical* for psychological research (i.e., d USA = 0.35), varying cultural influence. We ran 3000 simulations, 1000 with no cultural influence (0), 1000 with moderate cultural influence (0.5) and 1000 with strong cultural influence (1), sampling from 60 sites ($k = 60$).

2. In the second simulation, we again simulate a single study on a psychological effect across countries and estimate the *power to detect heterogeneity across different effect sizes*, varying d USA from d USA = 0.05 to d USA = 0.7, as well as varying *cultural influence*. We ran 2000 simulations, 1000 with moderate cultural influence (0.5) and 1000 with strong cultural influence (1), sampling from 60 sites ($k = 60$).

3. In the third simulation, we simulate an ML2-like project comprising several studies used for simulations 1 and 2, selecting d USA from the original results of ML2 and varying cultural influence. This allows us to estimate effective project-wide power to detect cultural heterogeneity. We ran 1000 simulations for moderate (i.e., 0.5) and strong (1) levels of cultural influence (2000 simulations in total), simulating a single social psychological effect in each simulation, with each "true" effect size being selected from a uniform distribution between d = 0.05 and d = 0.7, sampling from 60 sites ($k = 60$). A full description of the functions used in the simulation can be found in Supplementary

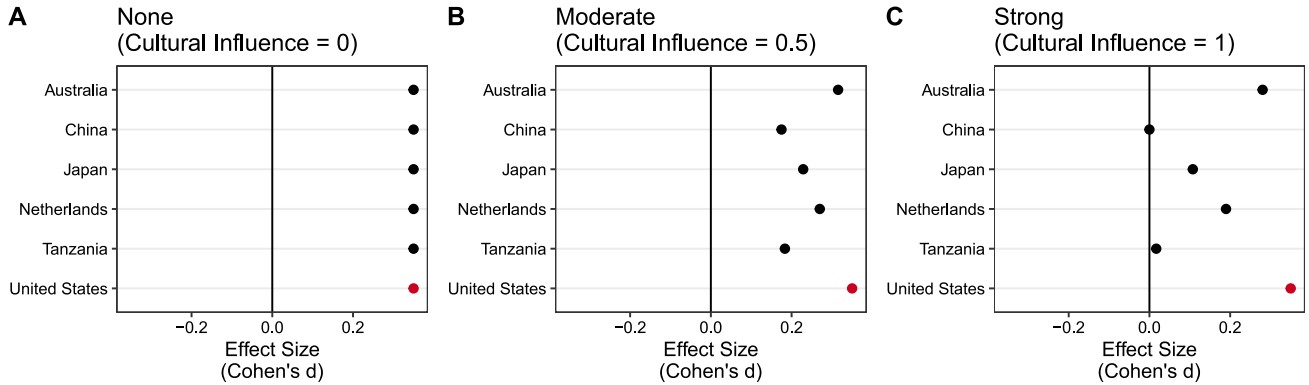

**Fig. 2 | The effect of culture on effect sizes at different levels of cultural influence.** Figure **A** shows Cultural Influence = 0 which means no cultural influence, Fig. **B** shows Cultural Influence = 0.5 which means moderate cultural influence, and Fig. **C** shows Cultural Influence = 1 which means strong cultural influence.

Section 1.1.1. The software used to write the simulation can be found in Supplementary Section 1.1.2.

We developed an alternative approach to analyze the ML2 data that addresses our stated Problems (3) *Conflating cultural background with sample country*, (4) *the ML2-WEIRDness variable*, and (5) *the Mean Split*. Unfortunately, neither an inductive measure for cultural differences nor a more accurate approach to identifying cultural background on the individual level can solve Problems (1) Lack of theory in the selection of studies and sample sites for replication, and (2) Sampling WEIRD people around the world. Moreover, as ML2 found that many of the results of the individual studies do not have evidentiary value at all, then we should not expect these null findings to vary cross-culturally. So, at best, an improved reanalysis will extract the variation in this available data, while recognizing that these samples may lack sufficient variation and statistical power, the protocol was implemented in a way that may have introduced noise, and the chosen studies only represent a subset of psychological science. To inform future studies in this realm, we chose to illustrate how some of the problems we state can be operationalized in an improved and theory-driven methodological approach.

We analyzed the existing ML2 data per the pre-registered analysis plan, registered on 18th January 2021 and available at https://doi.org/10.17605/OSF.IO/QRDXC. The analysis plan is summarized in Supplementary Fig. 2. All code used for analysis is available at https://doi.org/10.17605/OSF.IO/QRDXC. The software used to write the simulation can be found in Supplementary Section 1.2.3.

1. Specifically, our re-analysis incorporates three changes, compared to the ML2 approach, that may be useful for future studies to consider: We replaced the dichotomous *ML2-WEIRDness* score with a continuous proxy for cultural variation—cultural distance from the United States[10].
2. We operationalized cultural distance from the US not only at the sample site level but also at an individual level by identifying the participants' birth countries.
3. We calculated the cultural distance between participants (not just the distance to the US) based on their birth country. We used these between-level distances to estimate the moderating role of culture.
4. We explored whether the results for people who are native to a sample country are different from those in the same sample country who were born somewhere else (and may thus be described as having a different cultural background).

We detail the specific analysis method used to calculate the effect sizes for each study in Supplementary Table 2. We did not test whether the data met the assumptions of normal distribution or equal variance. For tests that assume normality, data distribution was assumed to be normal, but this was not formally tested. We did not request ethical approval for this study as LSE's statement on Research ethics states that "Research involving secondary analysis of established data sets from which it would not be possible to identify any living or recently deceased person need not be subject to the procedure, but wherever it is necessary for data to be effectively anonymised by LSE researchers, the procedure applies", and the ML2 data have already been anonymised.

**Cultural distance.** Muthukrishna et al.[10] developed a tool to measure the degree of similarity between the cultural values, beliefs, and practices of different groups of people. This measure provides a way to quantify worldwide variability in culture. For example, one can quantify the overall cultural distance between any two countries for which data is available (Muthukrishna et al.[10] used the 2005 and 2010 waves of the World Values Survey).

Compared to the pregistered approach, we used a version of $CF_{ST}$ based on a more recent version of the World Values Survey (WVS), which includes longitudinal data from the first 1981 wave until the 2017 wave (version "WVS_TimeSeries_1981_2020_spss_v2_0"), combined with the version used by Muthukrishna et al.[10]. When data for a country was available in the more recent version of the WVS, this data was used to calculate the $CF_{ST}$ for that country. If data were not available for a country in the more recent version, and data were available in the version used by Muthukrishna et al.[10], this data was used to calculate the $CF_{ST}$ for that country.

$CF_{ST}$ as implemented in our re-analysis focuses on the distance between each country and the United States, the WEIRD country par excellence that is vastly overrepresented in the psychological literature[8,10,33], as well as in ML2. Supplementary Fig. 1 shows the variation in distance from the United States in the ML2 data and its correlation to the ML2-WEIRDness scale. The samples come from a highly restricted range of countries, with 91% of the samples having a score of $CF_{ST} < 0.15$, even though world wide cultural distances from the United States extend to approximately 0.3 using data from the 2005 and 2010 World Values Survey data 10.

**Data exclusions.** We treated the raw data with the same exclusion protocol as the ManyLabs2 study (https://manylabsopenscience.github.io/ML2_data_cleaning). Furthermore, we employed further exclusion in either of the following two cases:

(1) Country of origin was based on participant's stated (1) hometown and (2) birth country. The percentage of individuals where their stated hometown is mismatched to their birth country is <1%. As we did not preregister to exclude these participants, they are included in the analysis. In cases (0.1%) where we were missing both participant's hometown and birth country, we ran two analyses: an analysis assuming they are native to the sample country and an analysis in which these participants are excluded.

Therefore, for analyses based on participants birth country/ hometown we will run three analysis variants where country of origin is based on participants:

- Birth country, excluding participants missing birth countries and hometowns (Birth country filtered)
- Birth country, where participants missing birth countries are assumed to be native to the sample country (Birth country imputed)
- Hometown

(2) There are some countries for which we do not have data for calculating cultural distance. In total 5/36 source countries were missing $CF_{ST}$: Austria, Belgium, Costa Rica, Portugal and the United Arab Emirates. Overall we have observations for 92.12% of the participants in the sample (including the USA). Therefore, we ran two versions of every model, one that excludes these countries, and another in which we estimate the cultural distance scores of these countries by averaging across the scores of their immediate geographic neighbors when those neighbors' scores are known (imputed $CF_{ST}$). A better analysis would be to also use other information, such as history, cultural phylogeny, language and/or institutional data to impute these values, but this would be a large project unto itself, so instead, we checked for robustness using an average of countries' nearest neighbors. The results of the analyses using imputed $CF_{ST}$ are reported in Supplementary Section 2.3.

**Sample sizes**. In our preregistration, we did not state our planned minimum sample size for inclusion. We chose to run the analyses with a number of different minimum sample sizes, ranging from stricter (100) to loose (10).

For the main analyses that we report in the body of the paper we used a minimum sample size of 36 participants per sampling site/country, as this was the minimum number of participants of a sample included in the original ManyLabs2 analysis (the uniporto sample from Portugal). We follow this threshold for convenience because the stricter criteria are even less tenable to illustrate the methodological approach because they would result in more exclusion of countries and raise issues with statistical power.

For the stricter analyses we used minimum sample sizes of 50 and 100 participants per sampling site/country. For the permissive analyses we used a minimum sample size of 10 participants per sampling site/country.

**Analysis A**. Does cultural distance measured by the Muthukrishna et al. cultural distance scale at a sample level explain variation in the outcomes of the studies? To ensure comparability, here we used an almost identical analytical approach as the authors of the ML2, except that we replaced the dichotomous *ML2-WEIRDness* score with the continuous $CF_{ST}$-based cultural distance score[10]. As such, we ran a Random-Effects model with cultural distance as a moderator, and similarly established heterogeneity of samples, using the Q, tau, and $I^2$ measures[16]. The specific variants of models used in this analysis are reported in Supplementary Section 1.2.2.1.

**Analysis B**. Does cultural variation identified on the individual level by the Muthukrishna et al. WEIRD scale explain variation in the behavior of participants? In this analysis we used a multilevel model (MLM) to predict effect sizes. For this analysis, we relied on the variance of cultural distances to the US at the sample site level (Analysis B1) and in a second approach also on the individual level, identifying $CF_{ST}$ by the birth country of participants (Analysis B2). The specific variants of models used in this analysis are reported in Supplementary Section 1.2.2.2.

We had additionally preregistered that we would run an analysis where we would include the sample site of the individuals as a random effect as a robustness check. However, after applying the inclusion criteria, we were unable to run this analysis as there are a greater number of source sites than birth countries.

**Analysis C**. Does the cultural distance between participants at an individual level explain variation in the behavior of participants? For this

analysis, rather than the distance from the US, we used the direct difference in $CF_{ST}$ between countries. We used matrix regression models with the same samples as Analysis B1 and B2 to assess whether individual-level differences between participants explain variations in their behavior. The specific variants of models used in this analysis are reported in Supplementary Section 1.2.2.3.

**Analysis D**. For each study, we preregistered whether we expect culture to matter or not. See Supplementary Table 3 for which studies we hypothesized could vary culturally. We ran our analyses on this subset of culturally relevant studies as well as the full set of studies. We conducted Analyses A-C only on the subset identified in Supplementary Table 3. Here, we first included studies we expect to cross-culturally vary. Next, we also included studies that may vary with some caveats and go/no-go exclusion criteria described in the comments. We also ran an exploratory analysis on all studies that were replicated in ML2. For each study, we have stated whether we expect $CF_{ST}$ to matter or not ex-ante (see Supplementary Section 1.2.2.4).

**Analysis E**. How similar is the behavior of participants in their native country to participants not in their native country (i.e., how different are migrant populations in their country of origin)? We pre-registered to compare the behavior of the native and migrant participants for Analysis A-D. However, we were unable to run these analyses, as only the largest (and often WEIRDest) countries reached minimum sample sizes for inclusion (e.g., as per the above inclusion criteria of 36 participants). Therefore, to better understand whether migration status has some impact on behavior, we performed an explorative analysis of whether there were differences in behavior between migrant and non-migrant participants ignoring the potential effects of $CF_{ST}$. That is, we re-ran the first-stage ML2 regression models and simply added 'migration status' as a control variable. Participants were defined as having a migration status when their stated birth country differed from the source country site where data was collected. An obvious improvement to this analysis would be to adjust for the time spent in the home country and at what ages. We decided to focus on samples situated in the US for our explorative analysis, as it had the largest sample size among those countries and also serves as a reference country for the $CF_{ST}$. We included studies where we used either linear regression or logistic regression as the statistical analysis method. In total five studies were excluded from this analysis. Four studies were excluded as they had different experimental designs meaning that a regression model was inappropriate for re-analysis. For the full details of the excluded studies and the models used see Supplementary Section 1.2.2.5.

### Reporting summary
Further information on research design is available in the Nature Portfolio Reporting Summary linked to this article.

## Results
### Literature search
Our literature review reveals that from its publication in 2018, over 50 scientific articles have referred to cultural differencs, the moderating role of culture, or WEIRD, when citing Klein et al. Table 2 depicts the relative and absolute frequencies. Additionally, we found that papers directly adopted the *ML2-WEIRDness* scale in their research[27,28].

### Simulating the moderating role of culture results
**Power to detect heterogeneity for typical effect size and across different effect sizes**. First, we sought to assess how much power ML2 had to detect different levels of cultural influence given a typical effect size for social psychology. We simulated 3000 single multi-site style investigations of a typical social psychology effect, each investigating a "true" $d_{USA} = 0.35$ effect size, with $k = 60$ samples drawn from countries in proportion to their actual representation in ML2.

**Table 2 | Relative and absolute frequencies of coding events in the 410 coded articles**

| | Citation category: "Replicability" | Citation category: "Moderation of Replicability" | Citation category: "Moderation by Culture/WEIRD" | Citation category: "Other" |
|---|---|---|---|---|
| Relative Frequency | 0.67 | 0.16 | 0.12 | 0.20 |
| Absolute Frequency | 270 | 66 | 50 | 81 |

50 articles cited ML2 referring to the cultural moderation analysis. As some papers cited Klein et al. [15] several times, they might have had several coding events. For the relative and absolute frequencies, an entry was counted if at least one of the coding events was in the respective category. This explains why the cumulative frequencies are larger than the total amount of coded papers.

**Table 3 | The values in the table show the power of the statistical tests used in ML2 to detect cultural influence for different analytical approaches and levels of cultural influence**

| Level of cultural influence | Q test | tau | $I^2$ | ML2-WEIRDness moderation |
|---|---|---|---|---|
| None (0) | 0.00 | 0.04 | 0.01 | 0.01 |
| Moderate (0.5) | 0.01 | 0.1 | 0.04 | 0.04 |
| Strong (1) | 0.06 | 0.44 | 0.32 | 0.28 |
| *Values Observed in ML2* | *0.39* | *0.32* | *0.46* | *0.11* |

The last column, titled "ML2-WEIRDness moderation," shows the power of the moderation analysis. The last row shows the share of studies in which ML2 found significant levels of heterogeneity across settings and different tests. The power to detect cultural heterogeneity with the ML2 WEIRDness scale ranges from 0 to 28%, depending on the simulated level of cultural influence. In the ML2 study, the authors found 3/28 (11%) studies with significant moderation via ML2-WEIRDness.

Our 3000 simulations consisted of 1000 simulated multi-site studies of a $d_{USA} = 0.35$ effect at each of the three levels of cultural influence (i.e., 0, 0.5, 1). We could perform the same analyses reported in ML2 for each of these simulated multi-site studies. By simulating a single study (e.g., one of the 28 studies ML2 ran) run at multiple sites, we could directly assess the statistical power at which cultural influence could be detected, given known effect sizes and rates of cultural influence that we directly controlled.

Overall, using Many Labs 2's criteria as benchmarks, power to detect cultural moderation was quite low. More specifically, the results show that statistical power is low to detect cultural influence on individual effects. That is, power is just 28% to detect cultural influence via the moderation test of *ML2-WEIRDness*, even if the modeled cultural influence is strong (see Table 3). Put differently, the 3/28 studies (11%) found in ML2 to be moderated by *ML2-WEIRDness* would best map to a scenario in which cultural influence across all studies is at least moderate (comparing the 0.11 in the lower right corner, with other values in that column). See Table 3 or an overview of estimated statistical power across tests.

This simulation suggests that for the typical effect sizes, one encounters in social psychology, ML2 was underpowered to detect all but the strongest levels of cultural influence. Strikingly, the power to detect moderation in which effects entirely disappear in countries dissimilar to the USA *(strong influence)* ranged from 6% to a mere 44%, depending on which of ML2's chosen criteria one focuses on. For the heterogeneity analysis using the *Q*, tau, and $I^2$ measures[16], results like those in ML2 are consistent with at least moderate (or strong) levels of actual cultural influence.

Similarly to a 2-condition between-subjects experiment testing a typical social psychology effect size (*d* = 0.35) with only a dozen participants, Many Labs 2 may have had less than 8% power to detect heterogeneity for many of its effects (power to detect *d* = 0.35 with 6 participants per condition = 0.08; power to detect moderate cultural heterogeneity, where effects half attenuate, for *d* = 0.35, by *tau* criterion = 0.08).

Second, we investigated the effect of heterogeneity at a range of effect sizes for the reference country, the USA. We ran 1000 simulations for moderate (i.e., 0.5) and strong (1) levels of cultural influence, simulating a single social psychological effect in each simulation, with each "true" effect size being selected from a uniform distribution between *d* = 0.05 and *d* = 0.7.

This allowed us to infer power to detect heterogeneity (using ML2's chosen criteria) across different effect sizes, using sampling plans directly modeled on ML2 itself.

Figure 3 shows that these simulations show that for all criteria for detecting heterogeneity (*Q*, tau, $I^2$, and *ML2-WEIRDness* moderation), power to detect moderate cultural influence was poor for all but quite large effect sizes.

This allowed us to infer power to detect heterogeneity (using ML2's chosen criteria) across different effect sizes, using sampling plans directly modeled on ML2 itself.

**Effective project-wide power to detect (cultural) heterogeneity**
Our simulations – at the level of an entire ML2 project, with multiple studies nested within dozens of sites – show that it takes substantial levels of (cultural) heterogeneity to consistently yield results comparable to those observed in a study with a sample composition and size of ML2 (see Supplementary Fig. 3). ML2 found that 39% of effects yielded *Q-test* effects significant at the 0.001 level. This pattern of results was most consistent with very strong levels of heterogeneity, where effect sizes fully reversed in highly dissimilar countries. Here, we simulate the results of many ML2 style studies with fixed heterogeneity (in the previous section, we simulated a single ML2 style study and many ML2 style studies with heterogeneity conditional on effect size).

By the criterion of tau >0.1, Many Labs 2 found that 32% of studies had significant heterogeneity across settings. This pattern of results was most consistent with moderate-to-strong cultural heterogeneity (assuming culture is the only source of heterogeneity), where effect sizes are attenuated by 50–100% in highly dissimilar countries. Slightly under half of the studies (46%) exhibited heterogeneity in settings, as indexed by $I^2$ values whose lower bounds exceeded zero. This value was most likely in our simulations when actual cultural heterogeneity was strong. DemographicsIn the ML2 data, the average $CF_{ST}$ distance from the United States using participants as the unit of observation is 0.062; using sample sites as the unit of observation, like in ML2, it is 0.055 ($CF_{ST}$ distance was collected in 2022). This is smaller than the cultural distance between the United States and Germany (0.069).

The ML2 sample consisted of 15305 participants sampled from 125 sample sites across 36 countries. The ML2 researchers determined the cultural background of participants based on the country a participant was sampled from. Strikingly, though not noted by Klein et. al., the samples had significant shares of migrants (e.g., international students) at some sites (up to 61% in the UAE and 45% in Canada; see Supplementary Tables 7:8). Figure 4 shows the constitution of birth countries for participants in the different source countries. Strikingly, many participants from the US were born in other countries, indicating the possibility of cultural variation hidden by the approach taken by ML2. Overall, we observe 15.78% migrants in the dataset. Supplementary Fig. 4 shows that determining the cultural background based on the home country of the participants, would have changed the constitution distribution of present countries.

The acronym-based operationalization of cultural difference reveals a highly skewed distribution, in which the *ML2-WEIRD* sample sites were culturally very similar. In contrast, the non-*ML2-WEIRD* samples had larger variations. Figure 5a shows the distribution for *ML2-WEIRDness* and the applied mean split on the country level.

**Fig. 3 | The power detect heterogeneity for typical effect size and across different effect sizes.** Figure **A** shows the Q test observed power across effect sizes, Fig. **B** shows the tau ≥0.10 observed power across effect sizes, Fig. **C** shows I² observed power across effect sizes and Fig. **D** shows the ML2-WEIRDness moderation observed power across effect sizes.

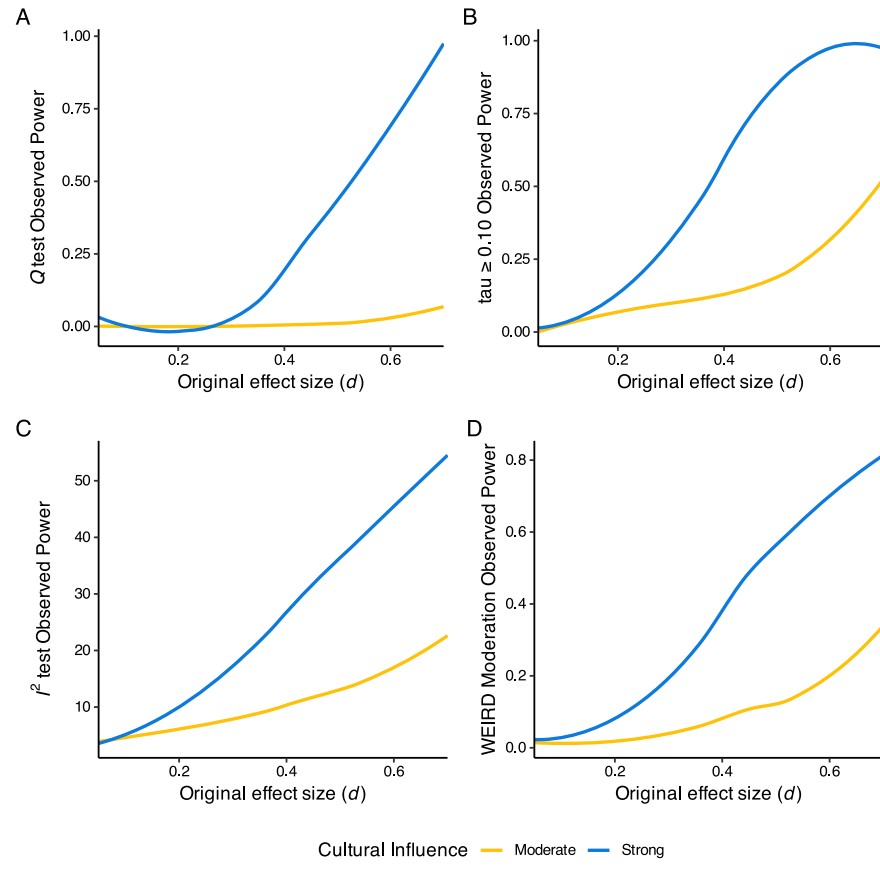

We report the main results of the re-analysis results in the main text, and the full results of all robustness tests can be found in Supplementary Section 2.3. In total, 14,096 participants were included in the Birth country filtered analysis, 14,220 participants were included in the birth country imputed analysis and 14,103 participants were included in the hometown analysis.

We found a significant level of cultural moderation in two of the four studies for which we predicted an effect because there is theoretical and empirical evidence for cultural variation (pre-registered). We also found evidence of cultural moderation in two other studies that did not have previous evidence for cross-cultural variation. A summary of the main results is presented in Table 4.

**Analysis A**. We hypothesized that the continuous $CF_{ST}$ cultural distance would be a stronger predictor of the effect size, compared to the ML2-WEIRDness score. Supplementary Fig. 5 summarizes the results in a Forest plot and shows that using $CF_{ST}$ instead of the binary ML2-WEIRDness variable only marginally increases the number of significant effects. Overall, $CF_{ST}$ consistently increases effect sizes for most effects (especially those effects that replicated in ML2; bolded in Supplementary Fig. 5) but likely failed to reach the chosen significance level because of constraints in statistical power. As per the simulation results above, the resulting pattern can be interpreted as preliminary evidence that moderate levels of cultural influence likely exist, but just replacing the binary ML2-WEIRDness variable with $CF_{ST}$ does not overcome the underlying issues in statistical power. The full results of all variants of Analysis A are presented in Supplementary Tables 9:11.

**Analysis B**. Analysis B1 shows a significant moderation of the $CF_{ST}$ by sample site for two effects, with only one of them having been successfully replicated in ML2 (see Supplementary Fig. 6). Analysis B2 shows no study being significantly moderated by $CF_{ST}$ by birth country (see Table 4).

Thus, the MLM regressions in Analysis B find less significant effects of $CF_{ST}$ than in Analysis A and the original analysis in ML2. There are several possible explanations for this result, including the type of models used or the relationship between cultural distance and the effect sizes not being linear. Due to our inclusion criteria of having at least 36 observations (the minimum sample size at ML2), many birth country sites were excluded from the analysis, which exacerbated the power issues. In total 1735 participants from 153 countries were excluded from analysis B2, removing much cultural variation (36 countries were retained in the analysis). We thus ended up with more sample locations than birth countries, which prevented us from running a pre-registered robustness check in which the sample sites of the individuals were included as a random effect. The full results of all variants of Analysis B are presented in Supplementary Tables 12:51.

**Analysis C**. Overall, we did not find any study with a consistently significant effect for $CF_{ST}$ in the matrix model (see Table 4). Similarly to analysis B, this is likely explained by constraints in the power, as both analyses use the same samples. The full results of all variants of Analysis C are presented in Supplementary Tables 52:83.

**Analysis D**. For 2/4 studies with theorical and empirical evidence for non-random cultural variation we found evidence of cultural variation in Analysis A (see Supplementary Table 84). For 2/6 studies with possible theorical and empirical evidence for non-random cultural variation we evidence of cultural variation in Analysis A. For 2/6 studies with theorical evidence for non-random cultural variation we found evidence of cultural variation in Analysis A. For 2/12 studies with possible theorical evidence for non-random cultural variation we found evidence of cultural variation in Analysis A. For Analysis B1 we found evidence of cultural variation in 1 study with theorical and empirical evidence for non-random cultural variation. For Analyses B2 and C we found no studies with evidence of cultural variation. Our results suggest that

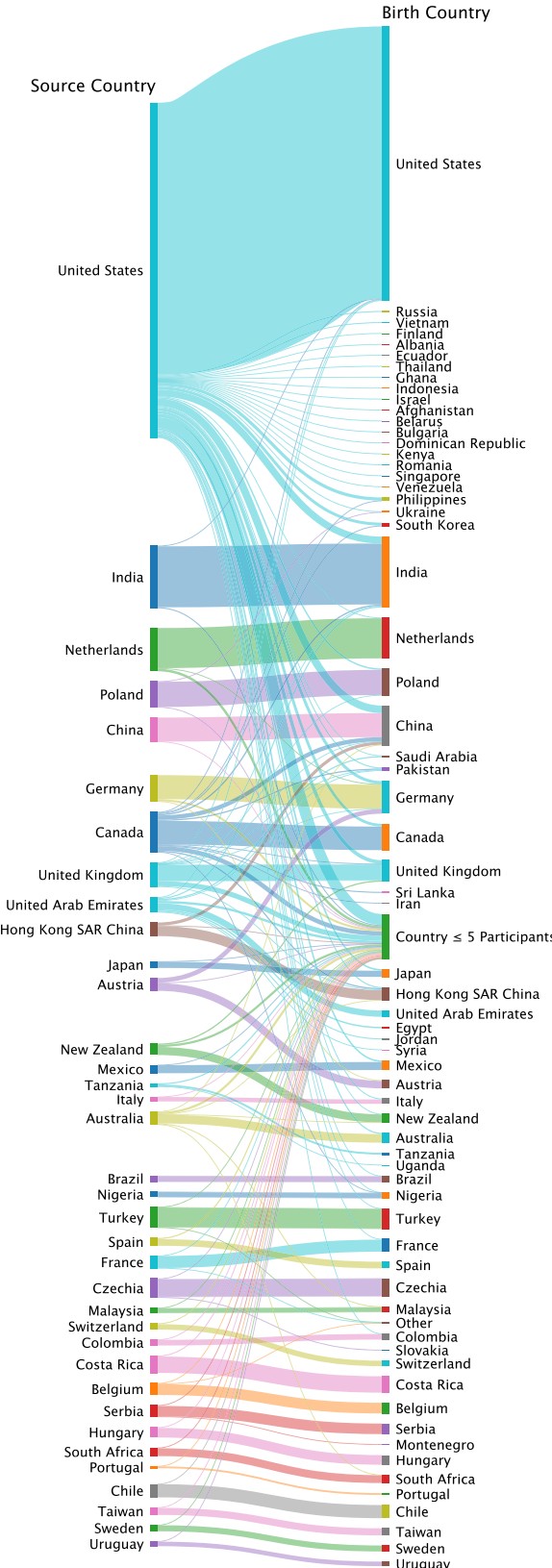

**Fig. 4 | The constitution of sample countries was based on birth countries as indicated by participants.** Source countries are shown on the left, and birth countries are on the right (A detailed overview including birth countries with less than 5 participants can be found in Supplementary Table 6).

Analyses B and C were not suitable to detect the moderating role of $CF_{ST}$ in the given setting.

**Analysis E**. Overall, we found that the migration status of participants had a significant moderation effect on the behavior in 12/23 of the studies included in the analysis (see Supplementary Fig. 7). The results of this exploratory, not pre-registered analysis cherry-picking one country should not be considered as an overall test of whether migration status matters but shows that classifying cultural background only based on the country where data is collected may neglect some rich cultural variation. The full results of all variants of Analysis E are presented in Supplementary Tables 85:87.

## Discussion

Here we synthesize the implications of the methodological problems, the simulation approach to detect statistical power, and the pre-registered re-analysis of ML2 to offer a set of guidelines and recommendations for more theoretically motivated, high-powered multi-site investigations of cultural differences in the future.

### Literature search

While the relative share of articles that cite ML2 for the cultural moderation analysis may appear small (12%), the absolute frequency (50) shows that this results from the large overall impact of the ML2 paper in the literature. This clearly shows that despite its exploratory nature, the conclusions drawn from this analysis have affected the psychological literature. Additionally, other research papers have directly adopted the *ML2-WEIRDness* scale[27,28]. Therefore, it is worth considering the appropriateness of this type of analysis when investigating cultural variability.

### Simulation

Our simulations show that for all criteria for detecting heterogeneity, power to detect moderate cultural influence was poor for all but quite large effect sizes. This combination of factors means that a multi-site investigation would have quite low power for detecting cultural heterogeneity for combinations of effect sizes and cultural influence that are quite plausible in the world (i.e., small-to-medium effects, with effect sizes largely attenuating in dissimilar populations). Because power to detect heterogeneity varied as a function of both the initial effect size and the degree of cultural influence – both quantities that one might hope to assess in a ML2-style project – it makes results from an ML2-style investigation difficult to interpret: is a given null result on a measure of heterogeneity reflective of an actual lack of heterogeneity, or merely low power to detect whatever amount of heterogeneity is present? Our simulation results indicate that the second scenario is plausible – a problem our later re-analysis also faces.

Our simulations show that, perhaps surprisingly, the observed rates of heterogeneity in ML2 are less consistent with simulations containing little actual heterogeneity and more consistent with strong heterogeneity that has not been accurately measured due to constrained statistical power. Observed rates of heterogeneity like those observed in ML2 only emerge in our simulations under conditions of very strong heterogeneity, where effect sizes either fully attenuate or flip. The sampling and analyses of ML2 may have given researchers only a small chance of detecting heterogeneity high enough that true effect sizes entirely reverse from country to country. Importantly, these findings do thus not imply potential issues in statistical power for the analysis considering culture, but the heterogeneity tests more broadly.

Despite all the limitations that apply to simulations in modeling real-world settings, these simulations give us some pause in evaluating ML2's conclusions. Across thousands of simulations in which culture mattered in degrees we could precisely control, analyses like those in ML2 usually failed to detect heterogeneity. Our findings suggest that like a 2-condition between-subjects experiment testing a typical social psychology effect size

**Fig. 5 | Frequency distribution for different measures of cultural differences.** Figure **A** shows the distribution of ML2-WEIRDness score in ML2 (data retrieved from Klein et al.[15] with cutoff for the binary mean split at 0.7) on the sample level, based on the source country. **B** Shows the distribution of $CF_{ST}$ values[10] on the sample level, based on the source country. **C** Shows $CF_{ST}$ distribution on the participant level, based on the source country, and **D** shows the $CF_{ST}$ values on the participant level, based on the birth country of participants. High values on the ML2-WEIRDness score indicate comparably "WEIRD" countries, while high values for $CF_{ST}$ indicate comparably "non-WEIRD" countries. As an illustrative example, the cultural distance between the United States and Germany is 0.069. The histograms show the data for Slate 1. A complete set of histograms, including data from Slate 2 can be found in Supplementary Fig. 4.

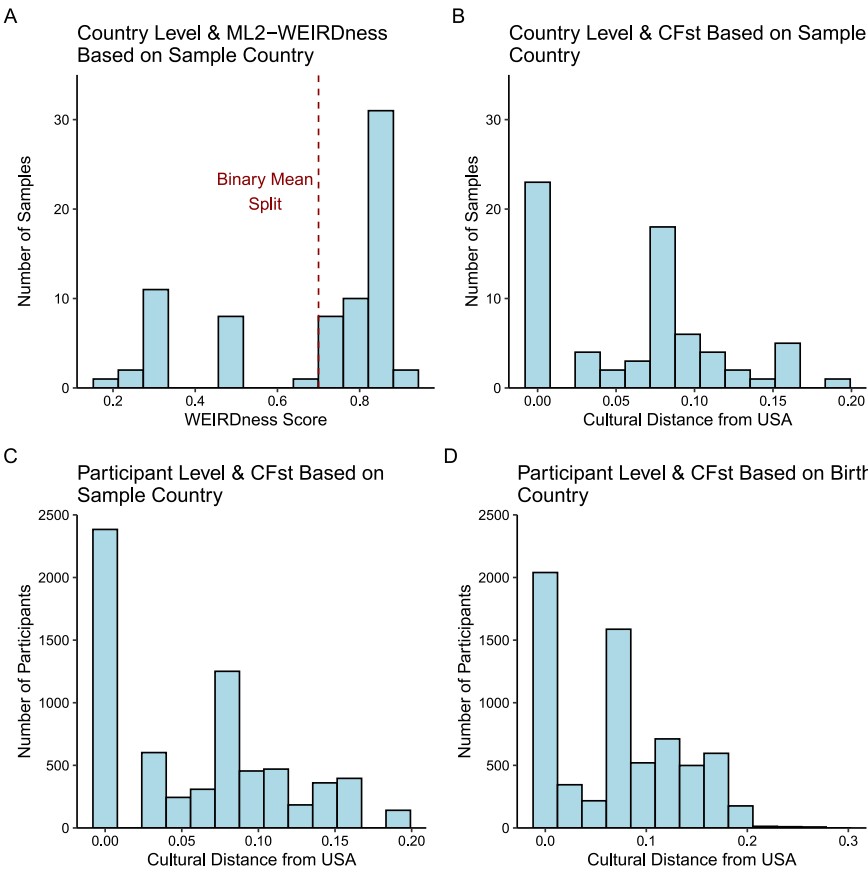

(*d* = 0.35) with only a dozen participants, Many Labs 2 might have had less than 8% power to detect heterogeneity for many of its effects. Put differently, any re-analysis of the data, including ours, will be similarly constrained in statistical power. Our simulations suggest that the methods used in ML2 are severely underpowered and thus preclude solid inference about the genuine degree of heterogeneity present.

Overall, other than our investigation of differences between native and migrant participants, our re-analysis did not change the substance of the results in ML2. Our chosen measure of cultural distance ($CF_{ST}$) consistently increases effect sizes for most effects (especially those effects that replicated in ML2; for effects that this does not replicate, we would have less reason to expect cultural moderation), but for many, it failed to reach the chosen significance level, potentially because of the constraints in statistical power. As per the simulation results above, the resulting pattern can be interpreted as preliminary evidence that moderate levels of cultural influence likely exist. However, just replacing the binary *ML2-WEIRDness* variable with another measure for cultural difference does not overcome the underlying issues in statistical power. The re-analysis does, however, show convincing evidence that the cultural variation in a given sample site is much larger if participants within a (sample) country are not lumped into the same cultural bracket but, for example, their migration history is respected when measuring cultural background (e.g., not coding European international students at US universities as culturally American).

## Limitations

Our pre-registered approach to improving the analytical protocol for detecting the impact of cultural distance from the US showed limited evidence that there may indeed be more cultural variance than detected in ML2. Based on the results from the simulation, the findings could be consistent with at least moderate levels of cultural influence in the ML2 sample.

However, this re-analysis cannot provide a satisfactory solution to measuring the moderating role of culture on replication success. That is, the re-analysis merely addresses one small aspect of the project – an issue with the analysis. The collected samples provide a comparably weak test of cross-cultural universality in effects for the abovementioned issues. Therefore, any analyses using these data are marred by the same serious design and sampling issues, which means that they cannot be the best test of the degree to which cultural differences matter for the selected psychological findings. Therefore, any attempts to improve any aspect of the analyses are limited by the range and structure of the data (e.g., identifying cultural background by home country potentially neglects rich regional diversity) and may, therefore, do little to change the results. Statistical analysis is a tool, and the raw materials one feeds a statistical model are at least as important as the modeling choices employed.

A more principled approach to testing the moderating role of cultural differences includes careful planning in both the design and analysis phases. In this paper, we addressed some important considerations for multi-site projects interested in cultural moderation (see ref. 34 for further discussion) and suggested practical ways to alleviate them. We hope that these above chapters prove to be practical guidelines for future studies in this realm. Nevertheless, improvements both at the design and the analysis stage are needed. Based on our above reasoning and drawing on arguments made in the literature, we suggest several points that need to be considered in the different stages of such a project.

## Culture may not moderate all aspects of psychology

The underlying theories should inform the selection of studies for a multi-site project tested[35]. Similarly, we suggest that effects for replication in a study testing the influence of sampling diversity should be selected based on theoretical predictions about whether we expect the effects to be moderated by culture. For example, for some effects, we would not expect successful replication across populations (e.g., North/South differences in

**Table 4 | A summary of the main results for analyses A-C, using a filtering criteria of a minimum 36 participants per country**

| Study | Original analysis | $CF_{ST}$ original analysis | $CF_{ST}$ source country | $CF_{ST}$ birth country (filter missing) | $CF_{ST}$ birth country (impute missing) | $CF_{ST}$ hometown country | Matrix model $CF_{ST}$ source country | Matrix model $CF_{ST}$ Birth Country (filter missing) | Matrix Model $CF_{ST}$ Birth Country (impute missing) | Matrix Model $CF_{ST}$ Hometown Country |
|---|---|---|---|---|---|---|---|---|---|---|
| Huang et al.[17] | 0.247* | −3.28* | −38.8* | −26.9 | −28.2 | −28.1 | −11.3 | −42.5 | −38.3 | −38.3 |
| Kay et al.[57] | 0.045 | −0.0557 | −0.0607 | −0.0626 | −0.0901 | −0.0798 | 0.205 | 0.193 | 0.241 | 0.23 |
| Alter et al.[58] | 0.0316 | −0.264 | −0.0123 | −0.067 | −0.0519 | −0.0663 | 0.122 | −0.426 | −0.345 | −0.428 |
| Graham et al.[54] | — | — | −1.55 | −1.51 | −1.64 | −1.6 | −4.55 | −0.282 | −0.315 | −0.239 |
| Rottenstreich and Hsee[59] | 0.0167 | −0.0951 | 0.0947 | −0.0681 | −0.203 | −0.0489 | 0.269 | −0.364 | 0.0457 | −0.387 |
| Bauer et al.[60] | −0.00584 | 0.00511 | −0.0664 | 0.0531 | 0.0477 | 0.0433 | 0.158 | −0.0952 | −0.205 | −0.144 |
| Miyamoto and Kitayama[61] | −0.0291 | 0.27 | 0.49 | 0.637 | −0.189 | 0.663 | 0.0896 | 1.65 | 3.85 | 1.47 |
| Inbar et al.[55] | — | — | 0.214 | 0.176 | 0.16 | 0.146 | −3.29 | −0.384 | −0.45 | −0.409 |
| Critcher et al.[62] | −0.00971 | 0.0304 | 0.527 | 2.33 | 2.07 | 2.28 | 0.195 | 2.67 | 0.408 | 2.21 |
| Van Lange et al.[63] | 0.0319 | 0.0753 | −0.936 | 0.0708 | 0.00926 | 0.0307 | 1.14 | −0.449 | −0.672 | −0.293 |
| Hauser et al.[64] | 0.0735 | −0.268 | 0.712 | −0.341 | −0.344 | −0.346 | −3.66 | 1.02 | 1.02 | 1.11 |
| Anderson et al.[65] | 0.0259 | 0.339 | −0.0172 | −0.0392 | 0.00892 | −0.0403 | −0.00178 | −0.0281 | 0.178 | −0.000667 |
| Ross et al.[65] | −0.0413 | 0.456 | −1.67 | −2.38 | −2.78 | −2.43 | −3.11 | −5.66 | −3.55 | −4.45 |
| Ross et al.[66] | −0.00127 | 0.0276 | 0.491 | 0.262 | 0.521 | 0.829 | −2.98 | −0.983 | −0.966 | −1.16 |
| Giessner and Schubert[67] | −0.00122 | −0.0367 | 0.0321 | −0.0387 | 0.0104 | 0.0362 | −0.0287 | 0.0921 | −0.0937 | 0.12 |
| Tversky and Kahneman[68] | 0.0453 | −0.27 | 0.0353 | −0.141 | −0.144 | −0.139 | −0.308 | −0.248 | −0.317 | −0.391 |
| Hauser et al.[64] | 0.0303 | −0.695* | 0.11 | 0.363 | 0.258 | 0.126 | −0.101 | 0.115 | 0.459 | 1.62 |
| Risen and Gilovich[69] | 0.0418 | −0.061 | −0.0154 | −0.13 | −0.268 | −0.154 | 0.00375 | 0.0893 | 0.917 | 0.031 |
| Savani et al.[70] | −0.0456 | 0.635* | −0.863 | 1.84 | 1.46 | 1.45 | 0.696 | 3.35 | 1.38 | 1.31 |
| Norenzayan et al.[19] | 0.166* | −0.811 | 3.43 | 5.83 | 6.79 | 6.79 | −2.37 | 1.37 | 6.09 | 7.11 |
| Hsee[71] | 0.0823 | −0.137 | 0.0104 | 0.189 | 0.168 | 0.203 | −0.152 | −0.199 | −0.249 | −0.105 |
| Gray and Wegner[72] | 0.0978 | −0.547 | 0.318 | 0.0685 | 0.132 | 0.118 | 0.223 | −0.228 | 0.211 | 0.165 |
| Zhong and Liljenquist[73] | −0.0377 | 0.00645 | 0.00113 | −0.1 | −0.0712 | −0.0822 | 0.171 | −0.133 | −0.169 | −0.15 |
| Schwarz et al.[56] | — | — | −0.419 | −0.415 | −0.0348 | −0.123 | 0.364 | −0.51 | 0.437 | 0.0103 |
| Shafir[74] | 0.0251 | −0.265 | −0.0523 | 0.17 | 0.139 | 0.128 | 0.331 | 0.0662 | −0.18 | −0.165 |
| Zaval et al.[75] | −0.0488 | 0.499 | 0.0202 | 0.0889 | 0.0742 | 0.07 | 0.0666 | −0.019 | −0.0422 | −0.0528 |
| Knobe[18] | 0.214* | −1.1* | 0.481* | 0.686 | 0.968 | 1.02 | −0.24 | 0.696 | 2.41 | 3.03 |
| Tversky and Gati[76] | −0.0227 | 0.0384 | 0.00294 | 0.0525 | 0.0247 | 0.0242 | −0.0888 | −0.0543 | −0.198 | −0.163 |

The significance threshold for the then migration status analysis uses the alpha = 0.004 (Slate 1) and alpha = 0.003 (Slate 2) as calculated in the original ML2 analysis using Bonferroni correction for multiple comparisons. Three studies were not included in analysis A, as they were not included in the original ML2-WEIRD analysis[54–56] Graham et al.[54–56], Inbar et al.[55], and Schwarz et al.[54–56].
*0.05.

socioeconomic status would not be expected to replicate across countries with different geographic patterns of wealth like Canada[17]). Other effects may have clear theoretical or empirical evidence suggesting less heterogeneity across populations (e.g., despite differences in social norms between countries, children tend to respond to some novel social norms across several societies[36]). Not all aspects of human psychology/behavior are equally likely to vary across populations[9]. To understand which effects may and may not be moderated by culture, we must invest more effort into developing better theories for human psychology and behavior.

### Selecting sample sites based on predictions of meaningful cultural differences

As a rule, authors should explicitly justify and defend how they selected their populations and state which factors apart from culture are likely to vary between those populations (e.g., wealth, climate, nutrition, geography, education, recent events, etc.). When placed in a global perspective, overall, there is scant cultural variability in the ML2 sample.

The strongest possible test of cultural variation of a particular phenomenon would require sampling from populations known to vary maximally on a theoretically relevant dimension[37]. Developing a theory *a priori* may not always be possible. Still, without a sound theory to explain the source of cross-cultural variation, it is difficult to know the range of cross-cultural psychological differences represented by these sites, and this necessarily weakens any conclusions that can be drawn about any particular effect's cultural variability. Future tests of cultural moderation should thus strongly consider grounding sample and effect selection in theory.

Sample sites should be selected based on a prediction about whether we would expect the cultural differences to produce significant psychological differences. We appreciate that coordination and availability constrain multi-site projects' ability to selectively target sample-specific sample sites. However, more statistical power in this setting does not necessarily require sampling more people from where they are available. Still, it could also mean deliberately sampling at different sample sites chosen to reflect theoretical expectations of relevant cultural differences. A more limited set of better, more theory-driven samples may increase statistical power and thus provide a stronger test of cultural heterogeneity than a convenience sample. For example, the current distribution of samples and participants in ML2 suggests that the researchers could have greatly reduced the participants at US sample sites and more evenly distributed sites between non-US Majority World countries without compromising statistical power. Indeed, this approach might have ironically boosted statistical power to detect heterogeneity across sites by reducing the degree to which USA data overwhelmed whatever signal of heterogeneity was present. Put differently, fewer but better samples could increase power. Future multi-site studies interested in the moderating role of culture should thus focus on the cultural, and not just geographical, variation of sampled participants.

### Drawing on simulations to determine the required sample size

During the design stage, simulations can help to understand the required sample size better, given the expected effect of cultural differences. We provide the code to a simulation that models the characteristics of the ML2 environment, or more broadly, of a multi-site project investigating how culture moderates behavior. Future studies can adapt this simulation to investigate which design features of cross-population multi-site studies (such as future Many Labs studies) would have the greatest power to detect cross-cultural heterogeneity in effects. As far as we can tell, ML2 did not run power simulations for the heterogeneity test, and our analyses indicate a lack of statistical power to detect all moderating effects of culture, even if they had been present. The lack of discussion of moderation tests' power may suggest that heterogeneity tests' power was assumed to be high because the overall power to detect main effects was high. We encourage researchers to reflect more explicitly on the power of statistical tests that drive key inferences and the statistical power of the main effects[32].

### Administering fewer studies per participant

The replication implementation featured a long list of studies presented simultaneously (ML2 had 28 effects overall - 13 in Slate 1 and 15 in Slate 2). The effects were bundled and sequentially administered to participants in an extensive protocol online and in the lab, an understandable choice given the project's scope. However, despite being randomized in order, assuming that any context-specific task could produce noisy responses in such a setting is reasonable. Exploring the effects of order provides only a partial solution since different sequences may affect different individuals and populations, resulting in greater measurement error. To estimate the reproducibility of particular studies, they should endeavor to provide a strong test of those effects by closely recreating the situation experienced by the original participants. Having participants participate in such a lengthy series of studies is at odds with how the original studies were conducted and is likely to result in them being less engaged with the studies, ultimately providing a weaker reproducibility test. Future studies may thus consider administering fewer studies per participant.

### Operationalization of cultural difference

Psychologists interested in addressing questions about cultural differences need to familiar themselves with the growing field of cultural evolution e.g. refs. 38,39. The field offers both fresh conceptualizations of culture and a variety of dynamic theories and helpful frameworks. From the vantage point of this field, the continuing tendency of psychologists to equate "country" and "culture" is a relic of European nationalism, which should be discarded immediately[8–10].

Nevertheless, measuring culture and cultural differences is not easy. Whatever conceptualization of culture is chosen, future studies should ensure that cultural differences are conceptualized in a careful, theory-driven approach. The ML2 approach to decompose the WEIRD backronym may seem practically plausible but is not informed by a theoretical understanding of cultural variation. Other approaches to operationalizing measures of cultural differences via $CF_{ST}$[10], tightness and looseness[30], or individualistic and collectivistic cultures[31] may provide better alternatives to capture population variability inductively and along continuous (rather than dichotomous) metrics. However, an important point is to avoid conflating cultural differences with cross-national differences. Some of the largest cultural differences are found when comparing state societies with non-state societies[40]. Culture is not just cross-national and not necessarily linear. Still, it is embedded in intersecting distributions of cultural traits within societies[5,41–43], geographical regions[22], religious differences[25], exposure to markets[40,44], social classes[41,45], ethnicities[46], kinship systems[21] and political orientations[47–49]. Alternative approaches, such as operationalizing cultural clusters by online behavior[50] should be considered.

### Beyond WEIRD samples

Lastly, biased participant samples are only part of the WEIRD people problem. Many fields of psychological and behavioral science are also heavily biased towards WEIRD topics, WEIRD researchers, and WEIRD institutions, which can further bias the types of questions researchers ask of their WEIRD samples (for a further theoretical and methodological critique, see refs. 3,51).

### Conclusion

Large-scale research efforts involving different research teams from around the globe are a critical part of advancing the field of psychological science in the future. They help address the replication crisis[2] and the WEIRD people problem[8,9], but not necessarily the problem in theory[5]. We applaud the authors of the Many Labs 2 study[15], for their efforts to contribute to the field with an ambitious research project that included participants living in 36 countries. Our admiration drives the critiques in this paper, and we hope it will help motivate the theoretical and methodological changes needed to test the moderating role of culture properly.

## Data availability

The data we used for analysis is secondary data that was kindly shared with us by the authors of the Many Labs 2 study. The full dataset contains personally identifiable information of participants and according to the authors of ML2[15] cannot be publicly shared beyond what is available at ML2's repository (https://osf.io/ux3eh/)[52]. The intermediate and final results of our analysis are available at https://doi.org/10.17605/OSF.IO/QRDXC[53].

## Code availability

We have shared our code at https://doi.org/10.17605/OSF.IO/QRDXC[53].

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

## Acknowledgements

We thank Agnieszka Sobola and Sudha Jagatheesh Jayanand for supporting the literature search. M.M. acknowledges support from the Canadian Institute for Advanced Research (CIFAR) grant no. CP22-005 and Templeton World Charity Foundation grant TWCF0612. R.S. acknowledges support from the Swiss National Science Foundation (grant no. 100018_185417/1). The funders had no role in study design, data collection and analysis, decision to publish or preparation of the manuscript.

## Author contributions

*Conceptualization:* Robin Schimmelpfennig, Will Gervais, Ara Norenzayan, Steven Heine, Joseph Henrich, Michael Muthukrishna; ***Formal Analysis:*** Rachel Spicer, Robin Schimmelpfennig, Cindel White; ***Simulation:*** Will Gervais; ***Writing – Original Draft Preparation:*** Robin Schimmelpfennig, Rachel Spicer, Cindel White, Will Gervais, Ara Norenzayan, Steven Heine, Joseph Henrich, Michael Muthukrishna; ***Writing – Review & Editing:*** Robin Schimmelpfennig, Rachel Spicer, Cindel White, Will Gervais, Ara Norenzayan, Steven Heine, Joseph Henrich, Michael Muthukrishna.

## Competing interests

The authors declare no competing interests.
