## [Peer Review File - updated · Communications Psychology]

11th Sep 23

Dear Mr. Schimmelpfennig,

Thank you for your patience during the peer-review process. Your manuscript titled "A Problem in Theory and More: Measuring the Moderating Role of Culture in Many Labs 2" has now been seen by 2 reviewers, whose comments are appended below. You will see that they find your work of some potential interest. However, they have raised quite substantial concerns that must be addressed. In light of these comments, we cannot accept the manuscript for publication, but would be interested in considering a revised version that fully addresses these serious concerns.

We hope you will find the Reviewers' comments useful as you decide how to proceed. Should additional work allow you to address these criticisms, we would be happy to look at a substantially revised manuscript. If you choose to take up this option, please highlight all changes in the manuscript text file, and provide a detailed point-by-point reply to the reviewers.

Editorially, we consider it critical that you revise the introduction to transparently present the stated aims and interpretation of the findings in the original study, Many Labs 2 ("ML2"). As presented currently, the reader has the impression that the WEIRD finding was the central aspect of the original study, and that the original authors presented their findings as strong evidence in support of the WEIRD claim. As Reviewer 2 points out, the analyses related to WEIRD-ness were an exploratory analysis that was not the main focus of the paper, although it is mentioned in the Abstract. One key consideration for replication and re-analysis papers in Communications Psychology is whether and how past studies (still) influence the literature. As we understand it, your argumentation is that other researchers have overinterpreted the ML2 study findings. If the study is frequently cited as evidence for cross-cultural difference (or lack thereof), despite the authors' caveated presentation, this provides clear justification for a re-analysis of these findings. In the revision we would like to see an analyses of the body of literature that cites the original study that demonstrates a focus by subsequent researchers on the findings on heterogeneity in the original paper and that this finding has been overinterpreted. Please use an indexing database (such as pubmed or web of science) to select the body of literature (e.g., there are currently 410 publications citing ML2 in Web of Science). At the same time, please adopt a more matter-of-fact and tone towards the ML2 study and avoid what can be seen as a mis-characterization of the authors' aims and interpretation.

We also agree with Reviewer 1 that there is value in the power simulation and the development of a tool to help improve future studies. We encourage you to make this the focus of the manuscript and to provide evidence from other recent multi-national studies to demonstrate whether your approach confirms suitability of the respective designs.

Per our policy regarding making all data available for reviewers if requested, please provide access to the data and code necessary for the reviewers to replicate your analysis with your revisions.

If the revision process takes significantly longer than five months, we will be happy to reconsider your paper at a later date, provided it still presents a significant contribution to the literature at that stage.

Please use the following link to submit your revised manuscript, point-by-point response to the Reviewers' comments with a list of your changes to the manuscript text (which should be in a separate document to any cover letter) and any completed checklist:

[link redacted]

Please do not hesitate to contact me if you have any questions or would like to discuss the required revisions further. Thank you for the opportunity to review your work.

Best regards,

Jennifer Bellingtier

Jennifer Bellingtier, PhD

Senior Editor

Communications Psychology

EDITORIAL POLICIES AND FORMATTING

Editorial Policy: Policy requirements (Download the link to your computer as a PDF.)

Furthermore, please align your manuscript with our format requirements, which are summarized on the following checklist:

Communications Psychology formatting checklist

and also in our style and formatting guide Communications Psychology formatting guide .

* **CODE AVAILABILITY:** All Communications Psychology manuscripts must include a section titled "Code Availability" at the end of the methods section. In the event of publication, we require that the custom analysis code supporting your conclusions is made available in a publicly accessible repository; please choose a repository that provides a DOI for the code; the link to the repository and the DOI must be included in the Code Availability statement. Publication as Supplementary Information will not suffice. We ask you to prepare and upload code at this stage, to avoid delays later on in the process.

* **DATA AVAILABILITY:**

All Communications Psychology research manuscripts must include a section titled "Data Availability" at the end of the Methods section or main text (if no Methods). More information on this policy, is available at <http://www.nature.com/authors/policies/data/data-availability-statements-data-citations.pdf>.

At a minimum the Data availability statement must explain how the data can be obtained and whether there are any restrictions on data sharing. Communications Psychology strongly endorses open sharing of data. If you do make your data openly available, please include in the statement:

We recommend submitting the data to discipline-specific, community-recognized repositories, where possible and a list of recommended repositories is provided at <http://www.nature.com/sdata/policies/repositories>.

If a community resource is unavailable, data can be submitted to generalist repositories such as figshare or Dryad Digital Repository. Please provide a unique identifier for the data (for example a DOI or a permanent URL) in the data availability statement, if possible. If the repository does not provide identifiers, we encourage authors to supply the search terms that will return the data. For data that have been obtained from publicly available sources, please provide a URL and the specific data product name in the data availability statement. Data with a DOI should be further cited in the methods reference section.

REVIEWER EXPERTISE:

Reviewer #1 Cross-cultural, open science

Reviewer #2 social psychology, modelling

Reviewer #1 (Remarks to the Author):

Here is an excellent reanalysis of the ML2 project which, in my opinion, clearly demonstrates that WEIRD-ness (as backronym) could easily be associated with experimental outcomes, but that the original study was unable to detect this effect due to lack of power. They do this with compelling simulations of data and an easy to read table summarizing their results compared to the original results (Table 1). This in itself should be published.

The authors spend far less time in the paper discussing this important finding, and instead try to (re-)introduce a new measure of culture that they suggest in lieu of the categorization of WEIRD by the original ML2 study. Parts of this discussion I am wholly supportive of. Namely that WEIRD does not make a very good dichotomous variable because it ends up sorting countries into extremes when they are somewhere in the middle in terms of WEIRD-ness (like Chile). Thus, the idea of possibly having a continuous measure of WEIRD-ness is sound and would follow psychometric logic that things relating to culture are rarely dichotomous. This also follows a political economy perspective where Education, Industrialization and Democracy are certainly not dichotomous, and arguably Western and Rich might not be either. Also, they consider that background of each participant in a given experiment is important, possibly more important than the country in which the experiment takes place, given that many are immigrants. Excellent points.

However, they propose to measure culture using a cultural measure that was developed by some of the authors of the paper (Muthukrishna et al). This measure uses World Values Survey data among other things to generate cultural (dis)similarity scores that can be rotated to form distances from a given point (usually a country). In this case they make their cultural scores based on distance from the USA. They then go on to show slightly more findings in favor of a cultural effect, but still not much of an effect. I have some issues with this strategy.

1. This isolates WEIRD-ness as a cultural phenomenon. This is not exactly correct. Look at the acronym. Actually, it has almost nothing to do with culture, rather socio-economic development

and political systems. In fairness, many studies, including the original study, discuss WEIRD-ness from a cultural and behavioral perspective, and therefore it is not 'off the mark' to focus on culture here. If the authors are serious about reanalysis, they should also consider how results compare between using culture as moderator and other socio-economic variables. Maybe it is not culture at all that matters, but something else like incentive structures, etc.

2. This builds directly on the last point. This study makes a big deal about theory, and that the original study lacked theory. However, they also lack theory here. They simply take a cultural dissimilarity measure, and apply it to all experiments. But, this is atheoretical. Research on culture and personality for example, demonstrates that culture matters in some settings and not in others. So for example, a cultural measure of masculinity or conservative values might impact an experiment on political ideology and moral foundations (Nosek et al 2009), but maybe not consumerism (Bauer et al 2012). My point is that if the authors are going to push theory, then they should be able to develop cultural-specific theories for each experiment. The experiments test radically different things in many cases. I understand that the assumption was about WEIRD versus non-WEIRD samples in the original study, but I don't think any scholars are arguing for a standardized effect of WEIRD samples across all of personality and behavioral psychology... So if they are going to go so far as to develop a cultural theory of WEIRD-ness and suggest it could matter (if we had more power), then they should also link aspects of culture with each experiment. Like a different distance measure for each that relates to the cultural aspects of the experiment.

3. I understand why they took the US as the reference point, but why not vary this, or further test robustness? The cultural study of Muthukrishna et al also used China as a anchor for example.

4. The paper seems to be pushing the cultural scale of some of its authors. I think the scale is a useful and well-designed scale, but this stretches beyond the point of the paper which really is a demonstration that the original study's conclusion that WEIRDness doesn't matter is not justified. Is this just self promotion pushed into an otherwise sound study? I'm not sure...

I have some other concerns. The lack of replicability is one, and I'd like to see some of the missing files. I realize that data with identifying information cannot be shared, but the workflow of this paper is not computationally reproducible. I literally cannot check their results because I am missing the participant data. They could clean and anonymize the data so that only participant's country of birth appears in the analyses, or they could provide means, N and covariances to allow for analyses.

There are also some errors in the workflow I identified before giving up when I realized that it is not reproducible due to lack of data. Here are two in the `source_country.R` file:

1. The file “ML2_WEIRD_Nations.csv” does not exist in the original ML2 study’s nor in the current study’s repository. I found the average WEIRD score in the original study’s file “ML2_SourceInfoTable_Weird.csv”, and used this in my computational reproduction.

2. The variable “Country” is capitalized in the object weird_countries, but written in small in the join command (minor point, but still makes the script non-reproducible).

Given the crisis in science, I would prefer to computationally reproduce their findings before offer an accept recommendation.

Some other points:

They state: “Many Labs 2, designed to assess the role of sampling heterogeneity, appears to have made sampling choices by convenience.” It does not “appear” to be, but is a convenience sample. The original author’s methods are clear, they had an open call for collaborators across social media and their networks. They can take a firm stance here IMO.

“The selected effects or sample sites cannot be accounted for retrospectively.” I do not 100% agree. The original study did not plan on analyzing this, they realized that they could after (post hoc), and this seemed like a worthwhile endeavor. That means that they have not theoretically planned their sample, thus they also could not have theoretically hacked their sample to try and achieve certain results, it is just an exploratory analysis. I agree this is a weak test, but we could learn some grounded or descriptive information here, despite a lack of prereg or theory.

The strongest argument they make is that the sample was mostly WEIRD to begin with. I remember when I first saw the results of this project presented and was really shocked by the map and densities. Although they call it a ‘global collaboration’, it was not really. Without further statistical tests, 39% US and some kind of 70-80% WEIRD should is grounds to dismiss the findings of the original as an artifact of a poor sample and as pointed out here, lack of power.

Reviewer #2 (Remarks to the Author):

The paper under review, “A Problem in Theory and More: Measuring the Moderating Role of Culture in Many Labs 2” by Schimmelpfennig and colleagues, is framed as a critique of the 2018 report on the “Many Labs 2” (hereafter ML2) project by Klein and colleagues. While I had heard of the ML2 project, I had not previously read the paper on it. I have therefore read the ML2 paper alongside the paper under review. This has been a surprising experience. My impression is that the Schimmelpfennig and colleagues are attacking a straw man. I therefore start my review by a summary of my reading of the ML2 paper.

The ML2 paper reported preregistered replications of 28 published findings to examine variation in effect magnitudes across samples and settings. The organizers extended an open invitation to other researchers to participate in data collection. In the end, there were more than 60 sites attempting to replicate each effect. The distribution of sites seems to be a fair representation of the distribution of high-level research in experimental psychology: 38% in the US, Canada, Australia and New Zealand, 33% in Europe, 13% in South-East Asia, 7% in Latin America, and the remaining few percent scattered across a few countries in the Middle East and Africa. As the motivating question of ML2 is why replication attempts of psychological effects often fail, I think it is very reasonable that the study examined replication attempts that are representative of where research is typically conducted.

The focus of the ML2 paper is preregistered analyses of the replicability of each of the 28 published findings and the extent to which observed effect sizes varied across samples and settings. The conclusions from these analyses are that about half of the findings did not replicate in the pooled data; that effects that did replicate were typically smaller than in the original studies; that the standard deviation of effect sizes across sites was typically small, around 0.1 or less for all effects but one; and that the influence of task order and whether studies were delivered in lab or online was negligible. The main implication, as stated by the ML2 paper, is that when a finding does not replicate it is not sufficient to presume that moderating influences account for the observed variation in a phenomenon; moderation “is not a credible hypothesis until it survives a confirmatory test” and the paper notes that “investigating moderating influences is much harder than presently appreciated in practice”.

In addition to these preregistered analyses, the ML2 paper includes an exploratory analysis of the moderating influence of cultural differences between WEIRD and less WEIRD samples. This analysis constitutes a minuscule part of the paper. I found no explicit motivation for the inclusion of this analysis in the paper. However, a possible motivation can be inferred from the first mention of WEIRD in the paper: “The vast majority of the original studies were conducted in a Western, educated, industrialized, rich, democratic (i.e., WEIRD) society.” My impression is that the ML2 authors are suggesting that the effects they study may be especially likely to fail to replicate in societies that are culturally different from the original studies, but it’s a shame that they do not spell out what they are after. At any rate, the only conclusion drawn from this exploratory analysis is that

compelling evidence for differences between WEIRD and less WEIRD samples was only found for three out of 28 effects. Even this modest conclusion is followed up by an extensive disclaimer:

“However, our approach characterized cultural differences at the most general level possible—a dichotomy of WEIRDness—and ignored the rich diversity within that categorization. The distribution of WEIRDness scores was such that the WEIRD samples were highly similar in WEIRDness, and the less WEIRD samples varied substantially in WEIRDness. Figure 3 illustrates the highly skewed distribution. Countries with scores above 0.70 were categorized as WEIRD, and the rest were categorized as less WEIRD. Our summary analyses also did not address the possibility of highly specific regional variations, such as differences between U.S. and British samples, nor did they examine why differences were observed. Nor did these analyses investigate many interesting sampling moderators available in this data set, such as individual differences, gender, and ethnicity. Some moderating influences could be evaluated using the present data set; testing others will require new data collections. Also, a true examination of WEIRDness would need to more deliberately vary sampling across each of the WEIRD dimensions. Further analyses of the present data set may inspire hypotheses to be tested in future studies.”

I now turn to the paper under review. This paper is not concerned with the main findings of ML2 but only with the exploratory analysis of the moderating influence of cultural differences between WEIRD and less WEIRD samples. The authors discuss several concerns that they have, but I do not see that they affect the modest conclusion drawn by the ML2 paper. Instead, the concerns seem to say that the ML2 study fails as an attempt to determine the moderating influence of culture on psychological effects in general. But this failure is unsurprising as the ML2 was not designed, nor claimed, to make such an attempt.

For example, the first concern is that ML2 “did not explain their theoretical basis for selecting the studies to be replicated and did not provide theoretically-grounded predictions regarding which psychological effects should and shouldn’t generalize cross-culturally”. Sure. But that only reflects that this is not what the project was designed to do. The cultural aspect was an exploratory add-on and no general conclusions were drawn from its results.

The second concern is that the ML2 sample consisted mostly of WEIRD people. This is true, but as I discussed above, the collection of sites seems appropriate for what I perceive the ML2 project was intended to do. Moreover, it certainly contained a sufficient number of sites in less WEIRD societies to make statistically meaningful exploratory comparisons; this is evidenced by the fact that a significant difference was found for three effects for which cultural differences were expected by prior work. However, if the aim had instead been to determine the moderating influence of culture on psychological effects in general, everyone would agree that another sampling strategy would have been required. As the ML2 paper stated: “a true examination of WEIRDness would need to

more deliberately vary sampling across each of the WEIRD dimensions. Further analyses of the present data set may inspire hypotheses to be tested in future studies.”

Concerns 3-5 are about how the exploratory analysis could have been done differently, by taking participants country of birth into account, by using another measure of WEIRDness (a measure of the cultural difference to the US), and by using it as a continuous instead of a dichotomous measure. As my quote from the ML2 paper makes clear, the ML2 researchers welcome additional analyses by other researchers. When the authors of the paper under review conduct such analyses, however, they do not settle on one alternative that they think ML2 should have used; instead, their supplement includes more than 80 tables of results from different variants. Interestingly, the results are overall very similar to those in the ML2 paper: when ML2 did not find compelling evidence for moderation by WEIRDness for an effect, the alternative analyses typically did not find such evidence either.

The paper under review also includes a simulation study that purports to show that ML2 had low statistical power to detect cultural influences. The idea is simply this: If an effect size is quite small already in WEIRD countries, say 0.3, and through cultural influences it is only half as big (0.15) in less WEIRD countries, this is difficult to detect. This is correct and, indeed, it mirrors a point I noted that the ML2 paper itself makes: “investigating moderating influences is much harder than presently appreciated in practice”. If a study aims at detecting cultural influences that shift sizes of effects by small amounts, it would certainly need higher power. However, this was not the aim of ML2. Moreover, I am doubtful whether that would be a worthy aim for any future study. There are so many clear and important cultural differences for psychologists to study; why invest an enormous effort in detecting negligible differences?

The paper concludes with some general guidelines for how to design studies to test theoretical predictions about the moderating role of culture on psychological effects. I did not find these guidelines very original. After all, there are numerous studies that examine cultural variation in various specific psychological effects. These guidelines are written as if this field does not exist already.

Re: Revision COMMSPSYCHOL-23-0203, now entitled “Does Culture Moderate the Replicability of Psychological Phenomena? Considerations in Theory and Methodology”

Dear Reviewer 1,

My co-authors and I would like to thank you for your constructive and insightful comments on our submission, originally entitled “A Problem in Theory and More: Measuring the Moderating Role of Culture in Many Labs 2” to *Communications Psychology*. We have reframed and rewritten large parts of the paper in light of your comments. To reflect the new framing, specifically that we do not want to only focus on ML2, but instead on informing future research, we have changed the title to “Does Culture Moderate the Replicability of Psychological Phenomena? Considerations in Theory and Methodology.” The paper is vastly improved due to the many insights from you, the editor, and the other referee. We hope you will agree and support publication in the journal.

I respond to your comments below, and I respond to the editors and the other referee in separate letters. In both cases, I quote from the revision as appropriate for convenience.

Summary Comments

Here is an excellent re-analysis of the ML2 project which, in my opinion, clearly demonstrates that WEIRD-ness (as a backronym) could easily be associated with experimental outcomes, but that the original study was unable to detect this effect due to lack of power. They do this with compelling simulations of data and an easy to read table summarizing their results compared to the original results (Table 1). This in itself should be published. The authors spend far less time in the paper discussing this important finding, and instead try to (re-)introduce a new measure of culture that they suggest in lieu of the categorization of WEIRD by the original ML2 study. Parts of this discussion I am wholly supportive of. Namely that WEIRD does not make a very good dichotomous variable because it ends up sorting countries into extremes when they are somewhere in the middle in terms of WEIRD-ness (like Chile). Thus, the idea of possibly having a continuous measure of WEIRD-ness is sound and would follow psychometric logic that things relating to culture are rarely dichotomous. This also follows a political economy perspective where Education, Industrialization and Democracy are certainly not dichotomous, and arguably Western and Rich might not be either. Also, they consider that background of each participant in a given experiment is important, possibly more important than the country in which the experiment takes place, given that many are immigrants. Excellent points. However, they propose to measure culture using a cultural measure that was developed by some of the authors of the paper (Muthukrishna et al). This measure uses World Values Survey data among other things to generate cultural (dis)similarity scores that can be rotated to form distances from a given point (usually a country).

Thank you for the summary of the paper and its contributions. We agree with your overall assessment of our contributions and have thus increased the focus on the simulation in the paper. To address your main comments, we have extended the chapter with the simulation and moved part of the re-analysis to the Supplementary. Furthermore, to not overly focus on only one potential measure of culture (e.g., CFst), we have also discussed other approaches, like tightness looseness,¹ or individualism/collectivism.²

1. # Evaluation Comments

In this case they make their cultural scores based on distance from the USA. They then go on to show slightly more findings in favor of a cultural effect, but still not much of an effect. I have some issues with this strategy. This isolates WEIRD-ness as a cultural phenomenon. This is not exactly correct. Look at the acronym. Actually, it has almost nothing to do with culture, rather socio-economic development and political systems. In fairness, many studies, including the original study, discuss WEIRD-ness from a cultural and behavioral perspective, and therefore it is not 'off the mark' to focus on culture here. If the authors are serious about re-analysis, they should also consider how results compare between using culture as moderator and other socio-economic variables. Maybe it is not culture at all that matters, but something else like incentive structures, etc.

Thank you for this feedback. We agree that "WEIRDness" is, per se, not only a cultural measure. As you point out, it could be related to several factors associated with population variability. We chose to focus on culture and thus chose a measure associated with culture because this is what the Many Labs

2 project mentioned in connection to “WEIRDness” (see Abstract and Keywords). However, we agree that cultural variation is difficult to isolate by only looking at country-level variation. Other socioeconomic factors, environmental cues, and ecological patterns may matter. Furthermore, cultural variation does not follow country borders, and country-level measures of culture are always imperfect as a result. Consequently, we have rewritten the paper in several sections with your comments in mind. Most importantly, we have taken the subchapter about the CFst out of the main paper.

2. **This builds directly on the last point. This study makes a big deal about theory, and that the original study lacked theory. However, they also lack theory here. They simply take a cultural dissimilarity measure, and apply it to all experiments. But, this is atheoretical. Research on culture and personality for example, demonstrates that culture matters in some settings and not in others. So for example, a cultural measure of masculinity or conservative values might impact an experiment on political ideology and moral foundations (Nosek et al 2009), but maybe not consumerism (Bauer et al 2012). My point is that if the authors are going to push theory, then they should be able to develop cultural-specific theories for each experiment. The experiments test radically different things in many cases. I understand that the assumption was about WEIRD versus non-WEIRD samples in the original study, but I don't think any scholars are arguing for a standardized effect of WEIRD samples across all of personality and behavioral psychology. So if they are going to go so far as to develop a cultural theory of WEIRD-ness and suggest it could matter (if we had more power), then they should also link aspects of culture with each experiment. Like a different distance measure for each that relates to the cultural aspects of the experiment. I understand why they took the US as the reference point, but why not vary this, or further test robustness? The cultural study of Muthukrishna et al also used China as an anchor for example. The paper seems to be pushing the cultural scale of some of its authors. I think the scale is a useful and well-designed scale, but this stretches beyond the point of the paper which really is a demonstration that the original study's conclusion that WEIRDness doesn't matter is not justified. Is this just self promotion pushed into an otherwise sound study? I'm not sure...**

These are all relevant points. We agree that the US should not be taken as a sole reference point and that theory is needed to understand which effects are likely to generalize across cultures and which effects are moderated by cultural variation. This was, in fact, one of the main motivations, and we have made several changes in the manuscript that should reflect this.

Also, before we ran the re-analysis, we pre-registered predictions about which effects should be influenced by “culture” (see Appendix D), which we then tested with the CFst scale. Put differently, we tried our best in the scope of such a project and the existing data we had.

However, we were also aware of the constraints in our paper, mainly in the part of the re-analysis. Thus, we have moved part of the pre-registered re-analysis to the Appendix. We hope that this addresses some of the reviewer's comments.

3. **I have some other concerns. The lack of replicability is one, and I'd like to see some of the missing files. I realize that data with identifying information cannot be shared, but the workflow of this paper is not computationally reproducible. I literally cannot check their results because I am missing the participant data. They could clean and anonymize the data so that only participant's country of birth appears in the analyses, or they could provide means, N and covariances to allow for analyses. There are also some errors in the workflow I identified before giving up when I realized that it is not reproducible due to lack of data. Here are two in the source_country.R file:**

1. The file “ML2_WEIRD_Nations.csv” does not exist in the original ML2 study’s nor in the current study’s repository. I found the average WEIRD score in the original study’s file “ML2_SourceInfoTable_Weird.csv”, and used this in my computational reproduction.

2. The variable “Country” is capitalized in the object weird_countries, but written in small in the join command (minor point, but still makes the script non-reproducible). Given the crisis in science, I would prefer to computationally reproduce their findings before offer an accept recommendation.

Since we did not collect the original data, and the original authors had not shared participant-level data publicly, we had some unclarity as to whether we could share it. But, we have now contacted the original authors, and they agreed that we should share the data for the peer review process. Thank you for pointing this out. We also ensured that the analysis is now fully reproducible per the documents available on the OSF page.

4. They state: “Many Labs 2, designed to assess the role of sampling heterogeneity, appears to have made sampling choices by convenience.” It does not “appear” to be, but is a convenience sample. The original author’s methods are clear, they had an open call for collaborators across social media and their networks. They can take a firm stance here IMO.

We agree with your point and have changed the respective section based on your feedback.

5. “The selected effects or sample sites cannot be accounted for retrospectively.” I do not 100% agree. The original study did not plan on analyzing this, they realized that they could after (post hoc), and this seemed like a worthwhile endeavor. That means that they have not theoretically planned their sample, thus they also could not have theoretically hacked their sample to try and achieve certain results, it is just an exploratory analysis. I agree this is a weak test, but we could learn some grounded or descriptive information here, despite a lack of prereg or theory.

Overall, we agree with your assessment. This was an explorative, ex-post analysis, so a priori theory could not have been done. Only criticizing ML2 for this approach would be insufficient. However, we still think there is value in explaining the importance of theory when investigating the moderating role of culture. This will inform future studies, who may choose a similar approach as ML2 or even use their developed scale.

Furthermore, despite being labeled as exploratory, the authors kept the results of the “WEIRDness” analysis in the main paper and featured the results in the Abstract. Following the editor's recommendation, we analyzed all articles (403 when we checked) that had cited ML2. This analysis shows that 50 papers have since cited ML2 with reference to moderation by “culture” / “weird”). Furthermore, the “ML WEIRDness” scale has been directly copied in several papers published since.^{3,4} We have added this citation analysis to the main text of our paper. Thus, we think it is okay to point out potential problems with this analysis and when it comes to theory and how future research can draw on those suggested improvements.

6. **The strongest argument they make is that the sample was mostly WEIRD to begin with. I remember when I first saw the results of this project presented and was really shocked by the map and densities. Although they call it a ‘global collaboration’, it was not really. Without further statistical tests, 39% US and some kind of 70-80% WEIRD should is grounds to dismiss the findings of the original as an artifact of a poor sample and as pointed out here, lack of power.**

We completely agree.

Thank you

Thank you again for your constructive response to our original submission. As you hopefully can see, we considered the feedback, and we reworked large sections of the paper as a result. We hope you agree that the paper is much improved and that you will support its publication.

Bibliography

1. Gelfand MJ, Raver JL, Nishii L, et al. Differences between tight and loose cultures: a 33-nation study. *Science*. 2011;332(6033):1100-1104. doi:10/dzt3n2
2. Hofstede G. *Culture's Consequences: Comparing Values, Behaviors, Institutions and Organizations Across Nations*. SAGE; 2001.
3. Van Assche J, Swart H, Schmid K, et al. Intergroup contact is reliably associated with reduced prejudice, even in the face of group threat and discrimination. *American Psychologist*. 2023;78(6):761-774. doi:10.1037/amp0001144
4. Topal MA, Aktas BE, Basoglu S, Harma M. The mediator role of willingness to sacrifice in the association between socio-economic status and relationship satisfaction. *Curr Psychol*. 2024;43(10):9480-9484. doi:10.1007/s12144-023-05097-9

Re: Revision COMMSPSYCHOL-23-0203, now entitled “Does Culture Moderate the Replicability of Psychological Phenomena? Considerations in Theory and Methodology”

Dear Reviewer 2,

My co-authors and I would like to thank you for your constructive and insightful comments on our submission, originally entitled “A Problem in Theory and More: Measuring the Moderating Role of Culture in Many Labs 2” to *Communications Psychology*. We have reframed and rewritten large parts of the paper in light of your comments. To reflect the new framing, specifically that we do not want to only focus on ML2 but rather on informing future research, we have changed the title to “Does Culture Moderate the Replicability of Psychological Phenomena? Considerations in Theory and Methodology.” The paper has vastly improved as a result of the many insights you, the editor, and the other referee have provided. We hope you will agree and support publication in the journal.

I respond to your comments below, and I respond to the editors and the other referee in separate letters. In both cases, I quote from the revision as appropriate for convenience.

1. Summary comment

The paper under review, “A Problem in Theory and More: Measuring the Moderating Role of Culture in Many Labs 2” by Schimmelfennig and colleagues, is framed as a critique of the 2018 report on the “Many Labs 2” (hereafter ML2) project by Klein and colleagues. While I had heard of the ML2 project, I had not previously read the paper on it. I have therefore read the ML2 paper alongside the paper under review. This has been a surprising experience. My impression is that the Schimmelfennig and colleagues are attacking a straw man. I therefore start my review by a summary of my reading of the ML2 paper.

The ML2 paper reported preregistered replications of 28 published findings to examine variation in effect magnitudes across samples and settings. The organizers extended an open invitation to other researchers to participate in data collection. In the end, there were more than 60 sites attempting to replicate each effect. The distribution of sites seems to be a fair representation of the distribution of high-level research in experimental psychology: 38% in the US, Canada, Australia and New Zealand, 33% in Europe, 13% in South-East Asia, 7% in Latin America, and the remaining few percent scattered across a few countries in the Middle East and Africa. As the motivating question of ML2 is why replication attempts of psychological effects often fail, I think it is very reasonable that the study examined replication attempts that are representative of where research is typically conducted.

The focus of the ML2 paper is preregistered analyses of the replicability of each of the 28 published findings and the extent to which observed effect sizes varied across samples and settings. The conclusions from these analyses are that about half of the findings did not replicate in the pooled data; that effects that did replicate were typically smaller than in the original studies; that the standard deviation of effect sizes across sites was typically small, around 0.1 or less for all effects but one; and that the influence of task order and whether studies were delivered in lab or online was negligible. The main implication, as stated by the ML2 paper, is that when a finding does not replicate it is not sufficient to presume that moderating influences account for the observed variation in a phenomenon; moderation “is not a credible hypothesis until it survives a confirmatory test” and the paper notes that “investigating moderating influences is much harder than presently appreciated in practice”.

We agree with your overall summary of the main goals of the ML2 project. We admit that in our initial submission, we may have treated the culture analysis as the main focus. We have thus reframed the introduction and tried to make it clearer that we focus on that part and not the whole paper. Thank you for pointing this out.

In addition to these preregistered analyses, the ML2 paper includes an exploratory analysis of the moderating influence of cultural differences between WEIRD and less WEIRD samples. This analysis constitutes a minuscule part of the paper. I found no explicit motivation for the inclusion of this analysis in the paper. However, a possible motivation can be inferred from the first mention of WEIRD in the paper: “The vast majority of the original studies were conducted in a Western, educated, industrialized, rich, democratic (i.e., WEIRD) society.” My impression is that the ML2 authors are suggesting that the effects they study may be especially likely to fail to replicate in societies that are culturally different from the original studies, but it’s a shame that they do not spell out what they are after. At any rate, the only conclusion drawn from this exploratory analysis is that compelling evidence for differences between WEIRD and less

WEIRD samples was only found for three out of 28 effects. Even this modest conclusion is followed up by an extensive disclaimer:

“However, our approach characterized cultural differences at the most general level possible—a dichotomy of WEIRDness—and ignored the rich diversity within that categorization. The distribution of WEIRDness scores was such that the WEIRD samples were highly similar in WEIRDness, and the less WEIRD samples varied substantially in WEIRDness. Figure 3 illustrates the highly skewed distribution. Countries with scores above 0.70 were categorized as WEIRD, and the rest were categorized as less WEIRD. Our summary analyses also did not address the possibility of highly specific regional variations, such as differences between U.S. and British samples, nor did they examine why differences were observed. Nor did these analyses investigate many interesting sampling moderators available in this data set, such as individual differences, gender, and ethnicity. Some moderating influences could be evaluated using the present data set; testing others will require new data collections. Also, a true examination of WEIRDness would need to more deliberately vary sampling across each of the WEIRD dimensions. Further analyses of the present data set may inspire hypotheses to be tested in future studies.”

We agree with your description of the approach of the cultural moderation analysis, which was, as you point out, explorative and not pre-registered. But we do disagree with the implications that you raise, namely that we are “attacking a straw man”. You argue that because it was “explorative”, “not explicitly motivated”, and “a minuscule part of the paper”, criticism is misguided. Again, we agree that the culture analysis was not the focus of the paper. But we disagree with this assessment that it is a neglectable, small part and, therefore, should not be criticized:

- *A small but relevant part of the paper:* Despite being labeled as exploratory, the authors presented the results of the “WEIRDness” analysis in the main paper, featured the results in the Abstract, and made “culture” one of the article’s keywords. In addition, the authors decided to feature this analysis in their widely shared social media coverage: “We also explored whether effects varied substantially between WEIRD and less WEIRD cultures. A couple of cases showed meaningful differences, but most did not. I find this Figure S2 to be particularly stunning. <https://psyarxiv.com/9654g>” (see Twitter link: <https://x.com/BrianNosek/status/1064551086210654209?s=20>)
- *Impact on the literature:* We conducted an analysis of all articles (403 when we checked), that had cited ML2. This analysis shows that 50 papers have since cited ML2 with reference to moderation by “culture” / “weird”). Furthermore, the “ML WEIRDness” scale has been directly copied in several papers that have been published since. ^{1,2} We have added this citation analysis to the main text of our paper.

Nevertheless, we do fully agree with your point that we need to do a better job of representing the goals of the ML2 project, and our initial submission has not achieved this. We have since made changes to all parts of the paper to better explain the main focus of the paper and to be less critical of ML2. We want this paper to help improve the design of future projects, and we are sure that the authors of ManyLabs 2 share this notion.

2. **I now turn to the paper under review. This paper is not concerned with the main findings of ML2 but only with the exploratory analysis of the moderating influence of cultural differences between WEIRD and less WEIRD samples. The authors discuss several concerns that they**

have, but I do not see that they affect the modest conclusion drawn by the ML2 paper. Instead, the concerns seem to say that the ML2 study fails as an attempt to determine the moderating influence of culture on psychological effects in general. But this failure is unsurprising as the ML2 was not designed, nor claimed, to make such an attempt.

For example, the first concern is that ML2 “did not explain their theoretical basis for selecting the studies to be replicated and did not provide theoretically-grounded predictions regarding which psychological effects should and shouldn’t generalize cross-culturally”. Sure. But that only reflects that this is not what the project was designed to do. The cultural aspect was an exploratory add-on and no general conclusions were drawn from its results.

The second concern is that the ML2 sample consisted mostly of WEIRD people. This is true, but as I discussed above, the collection of sites seems appropriate for what I perceive the ML2 project was intended to do. Moreover, it certainly contained a sufficient number of sites in less WEIRD societies to make statistically meaningful exploratory comparisons; this is evidenced by the fact that a significant difference was found for three effects for which cultural differences were expected by prior work. However, if the aim had instead been to determine the moderating influence of culture on psychological effects in general, everyone would agree that another sampling strategy would have been required. As the ML2 paper stated: “a true examination of WEIRDness would need to more deliberately vary sampling across each of the WEIRD dimensions. Further analyses of the present data set may inspire hypotheses to be tested in future studies.”

Concerns 3-5 are about how the exploratory analysis could have been done differently, by taking participants country of birth into account, by using another measure of WEIRDness (a measure of the cultural difference to the US), and by using it as a continuous instead of a dichotomous measure. As my quote from the ML2 paper makes clear, the ML2 researchers welcome additional analyses by other researchers. When the authors of the paper under review conduct such analyses, however, they do not settle on one alternative that they think ML2 should have used; instead, their supplement includes more than 80 tables of results from different variants. Interestingly, the results are overall very similar to those in the ML2 paper: when ML2 did not find compelling evidence for moderation by WEIRDness for an effect, the alternative analyses typically did not find such evidence either.

In the below paragraph, we directly quote from the ML2 paper and how they described their sample:

“Finally, we included an exploratory analysis of the moderating influence of cultural differences between WEIRD and less WEIRD samples. We sampled from 125 highly heterogeneous sources (39 U.S. samples and 86 samples from 35 other countries and territories) to maximize the possibility of observing variation in effects based on sample characteristics.” (p. 482)

Your interpretation of that the distribution of sites is “a fair representation of the distribution of high-level research in experimental psychology [...]”.

The distribution of sample countries does, per se, say little about the cultural variation. The overwhelming majority of participants came from the US and a handful of other countries (see Figure 1 in the main paper), and the cultural variation of the included countries was small (see Figure 3). You are probably right that this is a fair representation of subject pools in psychological research. However, it

seems an odd choice to test the potential influence of culture in a sample that has very reduced variation in culture. Your point may be that this is not what they set out to do. We agree, but the authors did present the results in the abstract in the following way: “Exploratory comparisons revealed little heterogeneity between Western, educated, industrialized, rich, and democratic (WEIRD) cultures and less WEIRD cultures (i.e., cultures with relatively high and low WEIRDness scores, respectively).” This is a fairly general statement about the differences between “WEIRD” and “less WEIRD” cultures. This was also perceived by many readers as such, as our citation review shows.

However, we agree that writing a paper that merely criticizes ML2 is not an important contribution to the literature. We have thus tried to rewrite the paper with a more nuanced approach. First, we want to provide some evidence of why we should be cautious in relying too much on the results from the cultural moderation test (which some people are doing, as our citation analysis shows). Second, we want to demonstrate why future studies should not take this same approach. Many Labs studies are seen by many researchers as the methodological gold standard (and we can see that several studies have already directly copied this approach), this is why we think it is ok to point out some shortcomings. Third, we want to provide concrete recommendations for how to improve the approach should researchers seek to run a similar study in the future.

3. **The paper under review also includes a simulation study that purports to show that ML2 had low statistical power to detect cultural influences. The idea is simply this: If an effect size is quite small already in WEIRD countries, say 0.3, and through cultural influences it is only half as big (0.15) in less WEIRD countries, this is difficult to detect. This is correct and, indeed, it mirrors a point I noted that the ML2 paper itself makes: “investigating moderating influences is much harder than presently appreciated in practice”. If a study aims at detecting cultural influences that shift sizes of effects by small amounts, it would certainly need higher power. However, this was not the aim of ML2. Moreover, I am doubtful whether that would be a worthy aim for any future study. There are so many clear and important cultural differences for psychologists to study; why invest an enormous effort in detecting negligible differences?**

The point of the simulation was to show that the results of findings in ML2 that show little effect for the moderating role of culture could have been because there is no cultural difference, which is a very important point for future multi-site studies. An alternative explanation could be that there are differences, but the paper did not have enough statistical power to detect those differences. This is a standard point in all empirical work, so it is unclear to us why it is not worth pointing out potential power issues within the heterogeneity test specifically because future studies will likely conduct similar heterogeneity tests and also fail to calculate the power at the right level.

In the introduction of the simulation chapter, we wrote:

“ML2’s moderation analysis “[...]revealed little heterogeneity between Western, educated, industrialized, rich, and democratic (WEIRD) cultures and less WEIRD cultures [...]”.^{3(p446)} This result can be interpreted in at least two ways. One possibility is that ML2 was well-designed, the analyses provide a strong test of cultural heterogeneity, and the results showing a small influence of culture are probably due to low rates of cultural heterogeneity. Another possibility is that culture has a large influence, and results indicate that the design and implementation of ML2 are poorly suited for detecting such differences. In essence, our simulation is trying to

understand which of the two scenarios (little heterogeneity, but well-measured; unknown-to-high heterogeneity, but poorly measured) is more likely to be true, given a design like MI.2.”

4. **The paper concludes with some general guidelines for how to design studies to test theoretical predictions about the moderating role of culture on psychological effects. I did not find these guidelines very original. After all, there are numerous studies that examine cultural variation in various specific psychological effects. These guidelines are written as if this field does not exist already.**

We have tried to better connect the paper to the existing literature, adding several citations. We have also tried to make it clearer that our guidelines are a synthesis of other recommendations that exist already.

Thank you

Thank you again for your insightful response to our original submission. As you hopefully can see, we considered the feedback, and we reworked large sections of the paper as a result. We hope you agree that the paper is much improved and that you will support its publication.

Bibliography

1. Van Assche J, Swart H, Schmid K, et al. Intergroup contact is reliably associated with reduced prejudice, even in the face of group threat and discrimination. *American Psychologist*. 2023;78(6):761-774. doi:10.1037/amp0001144
2. Topal MA, Aktas BE, Basoglu S, Harma M. The mediator role of willingness to sacrifice in the association between socio-economic status and relationship satisfaction. *Curr Psychol*. 2024;43(10):9480-9484. doi:10.1007/s12144-023-05097-9
3. Klein RA, Vianello M, Hasselman F, et al. Many Labs 2: Investigating Variation in Replicability Across Samples and Settings. *Advances in Methods and Practices in Psychological Science*. 2018;1(4):443-490. doi:10.1177/2515245918810225

26th Jul 24

Dear Mr Schimmelpfennig,

Your manuscript titled "Does Cultural Variation Moderate The Replicability of Psychological Phenomena? Considerations in Theory and Methodology" has now been seen by our reviewer, whose comments appear below. In light of their advice I am delighted to say that we are happy, in principle, to publish a suitably revised version in Communications Psychology under the open access CC BY license (Creative Commons Attribution v4.0 International License).

We therefore invite you to revise your paper one last time to address the remaining concerns of our reviewers and a list of editorial requests. At the same time we ask that you edit your manuscript to comply with our format requirements and to maximise the accessibility and therefore the impact of your work.

EDITORIAL REQUESTS:

SUBMISSION INFORMATION:

In order to accept your paper, we require only the files for the Manuscript, Supplemental Information (as PDF), Rebuttal letter, Cover Letter, Reporting Summary, and Editorial Policy Checklist.

OPEN ACCESS:

Communications Psychology is a fully open access journal. Articles are made freely accessible on publication under a CC BY license (Creative Commons Attribution 4.0 International License). This

license allows maximum dissemination and re-use of open access materials and is preferred by many research funding bodies.

For further information about article processing charges, open access funding, and advice and support from Nature Research, please visit <https://www.nature.com/commspsychol/article-processing-charges>

At acceptance, you will be provided with instructions for completing this CC BY license on behalf of all authors. This grants us the necessary permissions to publish your paper. Additionally, you will be asked to declare that all required third party permissions have been obtained, and to provide billing information in order to pay the article-processing charge (APC).

* **DATA AVAILABILITY:**

[link redacted]

Best regards,

Jennifer Bellingtier

Jennifer Bellingtier, PhD

Senior Editor

Communications Psychology

REVIEWERS' EXPERTISE:

Reviewer #1 Cross-cultural, open science

REVIEWERS' COMMENTS:

Reviewer #1 (Remarks to the Author):

The front end of the paper does justice to the original study and really focuses on the role of culture. Therefore, the criticism that follow now seem 'tempered' fairly. The materials have been mae

transparent or clarified. As far as I can tell, the authors have earned the right to present their work in published form.

Re: Revision COMMSPSYCHOL-23-0203, now entitled “Methodological Concerns Underlying a Lack of Evidence for Cultural Heterogeneity in the Replication of Psychological Effects”

Dear Editor,

Please find our response to your editorial requests below. We have restructured the manuscript into the format of ‘Introduction, Methods, Results, Discussion’ and believe our changes will fit with your style guidelines.

Move to Methods “We conducted a short systematic review of all published papers that cite the ML2 study to demonstrate this influence in the literature. Using Web of Science, we extracted all (403) research articles that cite Klein et al. 2018 (data extracted on September 18th, 2023).”

This section has been moved to the Methods and expanded upon to give the full details of our methodology.

Move to Results “This review reveals that from its publication in 2018, over 50 scientific articles have referred to the moderating role of culture when citing Klein et al. 2019. Table 1 depicts the relative and absolute frequencies.”

We have moved this section to the Results.

Move to Discussion “While the relative share of articles that cite ML2 for the cultural moderation analysis may appear small (12%), the absolute frequency (50) shows that this results from the large overall impact of the ML2 paper in the literature.”

We have moved this section to the Discussion.

Move to Results “Additionally, we found that papers directly adopted the ML2-WEIRDness scale in their research.”

We have moved this section to the Results.

Move to Discussion “This clearly shows that despite its exploratory nature, the conclusions drawn from this analysis have affected the psychological literature. Therefore, it is worth considering the appropriateness of this type of analysis when investigating cultural variability.”

We have moved this section to the Discussion.

Move to Results “Table 1. Relative and absolute frequencies of coding events in the 410 coded articles. 50 articles cited ML2 referring to the cultural moderation analysis. As some

papers cited Klein et al. 2018 several times, they might have had several coding events. For the relative and absolute frequencies, an entry was counted if at least one of the coding events was in the respective category. This explains why the cumulative frequencies are larger than the total amount of coded papers. For more info and a detailed explanation of coding, please refer to Supplementary Section 2.”

We have moved this section to the Results.

Hypotheses and research questions should appear at the end of the Introduction. “Here, we offer four contributions.”

We have moved our research questions to the end of the Introduction.

The concerns should all appear in the Introduction. The approach to validating these concerns should be presented in the Methods and the results of the analyses in the Results section. Discussion of the results and future directions should appear in the Discussion. I've further marked this in the subsequent section. “(4) treating the WEIRD backronym as a theory by decomposing it into a ML2-WEIRDness scale, and (5) using a mean split of that ML2-WEIRDness variable. Importantly, these points are all relevant for future cross-cultural multi-site studies that assess whether population variability moderates the replicability and size of psychological effects.”

We have restructured the manuscript so these parts are now correctly split into Introduction, Methods, Results and Discussion.

This does not need to be named as a research question as it is essentially the Discussion “We synthesize the implications of the methodological problems, the simulation approach to detect statistical power, and the pre-registered re-analysis of ML2 to offer a set of guidelines and recommendations for more theoretically motivated, high-powered multi-site investigations of cultural differences in the future.”

This point has been removed from the Introduction and put into the Discussion instead.

This section should follow under the Introduction heading. Sub-headings should be removed. “Theoretical and methodological considerations when testing the moderating role of culture”

Sub-headings have been removed from the Introduction.

Since our style does not allow for subheading in the Introduction, I recommend you start each of these sections in a systematic fashion, such as: Our first concern is X..., Our second concern is Y..., Our third concern is Z... “The importance of theory in the selection of studies and sample sites for replication.”

We have followed the suggested style and removed the subheadings from the Introduction.

Replace “It” with “They” in “It did not provide theoretically grounded (ex-post) predictions regarding which psychological effects will and will not generalize cross-culturally”

We have replaced this wording.

Our style does not allow for footnotes. Please incorporate into the main text or remove. “While Klein et al. did conduct subset analyses, including only participants from the US and Hong Kong respectively, for their main analysis they included the entire sample.”

This footnote has been incorporated into the main text.

Move to Discussion “As a rule, authors should explicitly justify and defend how they selected their populations and state which factors apart from culture are likely to vary between those populations (e.g., wealth, climate, nutrition, geography, education, recent events, etc.). The strongest possible test of cultural variation of a particular phenomenon would require sampling from populations known to vary maximally on a theoretically relevant dimension. Developing a theory a priori may not always be possible. Still, without a sound theory to explain the source of cross-cultural variation, it is difficult to know the range of cross-cultural psychological differences represented by these sites, and this necessarily weakens any conclusions that can be drawn about any particular effect’s cultural variability. Future tests of cultural moderation should thus strongly consider grounding sample and effect selection in theory”

This section has been moved to the Discussion.

Add citation or remove. “the WEIRDest of all countries”

We have added a citation and the qualifier, ‘arguably’.

More detail on how this was done should be supplied in the Methods “In the ML2 data, the average CF_{ST} distance from”

We have added a section to the Methods “Cultural Distance” which describes how CF_{ST} was calculated.

Report in the Results “using participants as the unit of observation is 0.062; using sample sites as the unit of observation, like in ML2, it is 0.055.”

This section has been moved to the Results.

Move to Discussion “Overall, there is scant cultural variability in the ML2 sample when placed in a global perspective.”

This section has been moved to the Discussion.

Move to Discussion “Future multi-site studies interested in the moderating role of culture should thus focus on the cultural, and not just geographical, variation of sampled participants”

This section has been moved to the Discussion.

Move to Results “Strikingly, though not noted by Klein et. al., the samples had significant shares of migrants (e.g., international students) at some sites (up to 61% in the UAE and 45% in Canada; see Tables S8: S9). Figure 2 shows the constitution of birth countries for participants in the different source countries. Strikingly, many participants from the US were born in other countries, indicating the possibility of cultural variation hidden by the approach taken by ML2.”

This section has been moved to the Results.

“Non-rich” non-WEIRD? Or do they specifically identify this aspect?

The UAE is specifically coded as non-rich in ML2, which is therefore the coding used for the American University of Sharjah sample - <https://osf.io/b7qrt>.

Move to Results “The acronym-based operationalization of cultural difference reveals a highly skewed distribution, in which the ML2-WEIRD sample sites were culturally very similar. In contrast, the non-ML2-WEIRD samples had larger variations. Figure 3a) shows the distribution for ML2-WEIRDness and the applied mean split on the country level”

“SIMULATING THE MODERATING ROLE OF CULTURE” This can become a subheading in the Methods and Results sections.

We have split this section into the Methods and Results as suggested.

“To calibrate expectations for what conclusions..” The first part of this section, up to the next subheading, should be part of the Introduction.

This section has been moved to the Introduction.

This section in Methods “Simulation Setup”

This section has been moved to the Methods.

Move to Results “Simulation Results Summary”

This section has been moved to the Results.

Move to Introduction. “Our initial simulation results focused on the power to detect different levels of cultural influence, given a single typical effect size. However, as Equation

1 shows, analytic performance can vary across both degrees of cultural influence and effect sizes. After all, ML2 included effect sizes ranging from practically nonexistent to quite large. Second, we investigated the effect of heterogeneity at a range of effect sizes for the reference country, the USA.”

This section has been moved to the Introduction.

Move to Methods “We ran 1000 simulations for moderate (i.e., 0.5) and strong (1) levels of cultural influence, simulating a single social psychological effect in each simulation, with each “true” effect size being selected from a uniform distribution between $d = .05$ and $d = .7$ ”

This section has been moved to the Methods.

Move to Results and add full results reporting “These simulations show that for all criteria for detecting heterogeneity (Q, tau, χ^2 , and ML2-WEIRDness moderation), power to detect moderate cultural influence was poor for all but quite large effect sizes.”

We have moved this section to the results and have added a figure showing the full results.

Move to Discussion “This combination of factors means that a multi-site investigation would have quite low power for detecting cultural heterogeneity for combinations of effect sizes and cultural influence that are quite plausible in the world (i.e., small-to-medium effects, with effect sizes largely attenuating in dissimilar populations). Because power to detect heterogeneity varied as a function of both the initial effect size and the degree of cultural influence – both quantities that one might hope to assess in a ML2-style project – it makes results from an ML2-style investigation difficult to interpret: is a given null result on a measure of heterogeneity reflective of an actual lack of heterogeneity, or merely low power to detect whatever amount of heterogeneity is present? Our simulation results indicate that the second scenario is plausible – a problem our later re-analysis also faces”

This section has been moved to the Discussion.

Needs to move to Introduction and some to Methods (with additional detail added) “In addition to running simulations at the level of a single study, we also tried to simulate the entire ML2 setup with several studies at a project level. We simulated a series of social psychological effects, with multiple studies run across countries, with effect sizes (dUSA) randomly drawn from those observed in ML2. We could then pool results across simulated slates of many studies to see which levels of simulated cultural heterogeneity would be expected to produce overall rates similar to those...”

We have split this section into the Introduction and the Methods and have added more details to the Methods.

Move to Results “Our simulations – at the level of an entire ML2 project, with multiple studies nested within dozens of sites – show that it takes substantial levels of (cultural)

heterogeneity to consistently yield results comparable to those observed in a study with a sample composition and size of ML2. ML2 found that 39% of effects yielded Q-test effects significant at the .001 level. This pattern of results was most consistent with very strong levels of heterogeneity, where effect sizes fully reversed in highly dissimilar countries. Here, we simulate the results of many ML2 style studies with fixed heterogeneity (in the previous section, we simulated a single ML2 style study and many ML2 style studies with heterogeneity conditional on effect size).

By the criterion of $\tau > 0.1$, Many Labs 2 found that 32% of studies had significant heterogeneity across settings. This pattern of results was most consistent with moderate-to-strong cultural heterogeneity (assuming culture is the only source of heterogeneity), where effect sizes are attenuated by 50-100% in highly dissimilar countries. Slightly under half of the studies (46%) exhibited heterogeneity in settings, as indexed by I2 values whose lower bounds exceeded zero. This value was most likely in our simulations when actual cultural heterogeneity was strong”

This section has been moved to the Results.

Move to Discussion “Our simulations show that, perhaps surprisingly, the observed rates of heterogeneity in ML2 are less consistent with simulations containing little actual heterogeneity and more consistent with strong heterogeneity that has not been accurately measured due to constrained statistical power. Observed...”

This section has been moved to the Discussion.

Please fully report all findings in the Results section “3 Power to detect $d = .35$ with 6 participants per condition = .08. Power to detect moderate cultural heterogeneity (effects half attenuate) for $d = .35$, by tau criterion = .08.”

These findings have now been reported in the Results section.

Analyses should be in Methods, and Results in Results

These sections have now been split.

Move to Introduction “With these observations and caveats out in front, we developed an alternative approach to analyze the ML2 data that addresses three of our stated considerations about the: (3) the cultural background of participants not necessarily being that of the sample country, (4) ML2-WEIRDness variable, and (5) Mean Split.

Unfortunately, neither an inductive measure for cultural differences nor a more accurate approach to identifying cultural background on the individual level can address considerations (1) the importance of theory in the selection of studies and sample sites for replication and (2) sampling mostly WEIRD people around the world. Moreover, as ML2 found that many of the results of the individual studies do not replicate and thus have no evidentiary value, we should not expect these null findings to vary cross-culturally. So, at

best, an improved re-analysis will extract the variation in this available data while recognizing that these samples may lack sufficient variation and statistical power. As previously stated in our pre-registration, the results of our re-analysis should thus not be perceived as a final verdict on the moderating role of culture in the given effects. Answering such a question about cultural heterogeneity would require data from a project specifically designed for the task. To inform future studies in this realm, we chose to illustrate how some of the problems we state can be operationalized in an improved and theory-driven methodological approach.”

This section has been moved to the Introduction.

Move to Methods “We analyzed the existing ML2 data per the pre-registered analysis plan described below. Specifically, our re-analysis incorporates three changes, compared to the ML2 approach, that may be useful for future studies to consider:”

This section has been moved to the Methods.

Move to Methods and ensure detail is sufficient for reproducibility “1. We replaced the dichotomous ML2-WEIRDness score with a continuous proxy for cultural variation—cultural distance from the United States. 10 2. We operationalized cultural distance from the US not only at the sample site level but also at an individual level by identifying the participants' birth countries. 3. We calculated the cultural distance between participants (not just the distance to the US) based on their birth country. We used these between-level distances to estimate the moderating role of culture. 4. We explored whether the results for people who are native to a sample country are different from those in the same sample country who were born somewhere else (and may thus be described as having a different cultural background).”

This section has been moved to Methods, and we have moved most of the methods from the Supplementary to the Methods section, to ensure that there is sufficient detail for reproducibility.

All main findings should be in the main Results section. “We report a summary of the re-analysis results in the main text, and the full results and documentation of this re-analysis can be found in Supplementary section three”

We can confirm that all the main results have been moved to the Results section.

Needs full statistical reporting in Results section “We found a significant level of cultural moderation in two of the four studies for which we predicted an effect because there is theoretical and empirical evidence for cultural variation (pre-registered). We also found evidence of cultural moderation in two other studies that did not have previous evidence for cross-cultural variation”

We have added Table 4 from the supplementary which summarizes the main results.

Move to Discussion “Overall, other than for point four, our re-analysis did not change the substance of the results in ML2. Our chosen measure of cultural distance (CFST) consistently increases effect sizes for most effects (especially those effects that replicated in ML2; for effects that this does not replicate, we would have...”

This section has been moved to the Discussion.

Place after References “Author contributions:”

This has been moved to after the references.

Place after References “Conflict of Interest:”

This has been moved to after the references.

Place after References “Acknowledgments:”

This has been moved to after the references.

Add to Acknowledgments and include role of funders. We do not allow a separate funding section. “Funding: M.M. acknowledges support from the Canadian Institute for Advanced Research (CIFAR) grant no. CP22-005 and Templeton World Charity Foundation grant TWCF0612. R.S. acknowledges support from the Swiss National Science Foundation (grant no. 100018_185417/1)”

The funding information has been added to acknowledgements and the role of funders has been stated.

Include in Methods with date of preregistration. “Pre-registration: <https://osf.io/qrdxc/>”

This has been moved to the Methods section with the date of preregistration.

Add to Methods “Supplemental Material: OSF Folder containing Pre-registration, Simulation Files, and Supplementary Information: <https://osf.io/qrdxc/>”

This section has been added to the Methods.

Include in Reference list “The ML2 Paper: <http://journals.sagepub.com/doi/10.1177/2515245918810225> ML2 OSF Folder: <https://osf.io/ux3eh/>”

These items have now been cited in the main reference list.

Create separate Code Availability statement. “We have shared our code and the intermediate and final results of our analysis at <https://osf.io/qrdxc/>.”

We have created a separate code availability statement.